# The Effects of Omega-3 Polyunsaturated Fatty Acids on Breast Cancer as a Preventive Measure or as an Adjunct to Conventional Treatments

**DOI:** 10.3390/nu15061310

**Published:** 2023-03-07

**Authors:** Matheus H. Theinel, Mariana P. Nucci, Arielly H. Alves, Olivia F. M. Dias, Javier B. Mamani, Murilo M. Garrigós, Fernando A. Oliveira, Gabriel N. A. Rego, Nicole M. E. Valle, Gabriela Cianciarullo, Lionel F. Gamarra

**Affiliations:** 1Hospital Israelita Albert Einstein, São Paulo 05652-000, Brazil; 2LIM44–Hospital das Clínicas da Faculdade Medicina da Universidade de São Paulo, São Paulo 05403-000, Brazil

**Keywords:** omega-3, breast cancer, preclinical, docosahexaenoic acid, polyunsaturated fatty acids, EPA, DHA

## Abstract

In order to understand how omega-3 polyunsaturated fatty acid (ω-3 PUFA) supplements affect breast cancer prevention and treatment, a systematic review of articles published in the last 5 years in two databases was performed. Of the 679 articles identified, only 27 were included and examined based on five topics, taking into account: the induction type of the breast cancer used in animal models; the characteristics of the induction model by cell transplantation; the experimental design of the ω-3 supplementation—combined or not with a treatment antitumor drug; the fatty acids (FAs) composition used; the analysis of the studies’ outcomes. There are diverse and well-established animal models of breast cancer in the literature, with very relevant histological and molecular similarities depending on the specific objective of the study, such as whether the method of tumor induction was transgenic, by cell transplantation, or by oncogenic drugs. The analyses of outcomes were mainly focused on monitoring tumor growth, body/tumor weight, and molecular, genetic, or histological analyses, and few studies evaluated latency, survival, or metastases. The best results occurred when supplementation with ω-3 PUFA was associated with antitumor drugs, especially in the analysis of metastases and volume/weight of tumors or when the supplementation was started early and maintained for a long time. However, the beneficial effect of ω-3 PUFA supplementation when not associated with an antitumor agent remains unclear.

## 1. Introduction

Breast cancer (BC) is the second most common neoplasm in the world, with 2.3 million new cases diagnosed in 2020. By 2040, due to population growth and aging, the worldwide incidence of breast cancer is expected to rise to more than three million cases with one million deaths each year [1]. BC normally affects young and mainly Black women, who represent 10% to 20% of BC cases. The risk of developing BC is influenced by factors such as lifestyle, physical activity, eating habits, genetic predisposition, sex, and age [2,3,4,5].

BC is widely characterized into four major subtypes, based on invasiveness, proliferation, and gene expression [6,7,8], known as (1) Luminal A, (2) Luminal B, (3) HER2+, and (4) Triple Negative. This classification of BC depends on the presence or absence of some biomarkers such as the estrogen receptor (ER), progesterone receptor (PgR), and human epidermal growth factor receptor 2 (HER2), which are used to diagnose and guide the adjuvant treatment options for BC.

In general, BC tumors classified as Luminal A are positive for ER and PgR and negative for HER2, exhibit slow growth, and have a good prognosis. Tumors classified as Luminal B, in turn, are positive for ER and negative for PgR and HER2, presenting a more accelerated growth and a less favorable prognosis. The HER2+ tumors only express the HER2 receptor, and, in spite of showing a faster growth, HER2-targeted therapies have achieved good response rates. Finally, tumors categorized as Triple-Negative BC (TNBC), which are negative for HER2, ER, and PgR, are usually associated with mutations in the breast cancer 1 (BRCA1) gene and represent the most aggressive and invasive BC type, with a mean of survival rate of 13 to 18 months after diagnosis [9].

The available treatments consist of partial or total mastectomy, chemotherapy, hormone therapy, immunotherapy, and/or radiotherapy [10,11]. When treated properly and in a timely manner, BC presents a good prognosis, in spite of chemo- or radiotherapy-related side effects that impact patients’ quality of life [12,13] and often lead to an increased treatment discontinuation rate [3,10,13]. It is crucial to develop ways in which to increase the efficacy of available medications and diminish the impact of the side effects; TNBC exhibits resistance to alternative treatments and does not show a significant response to targeted therapies [14,15,16].

In recent years, preclinical [17,18,19] and clinical studies [20,21] have shown that omega-3 polyunsaturated fatty acids (ω-3 PUFA) represent an efficient, supportive therapeutic approach for the treatment of breast cancer [22]. They are considered essential dietary nutrients considering that they cannot be produced naturally through biochemical and metabolic processes.

The use of ω-3 PUFAs, eicosapentaenoic acid (EPA), and docosahexaenoic acid (DHA) has been shown to minimize chemotherapy side effects and improve progression-free survival as well as the overall survival of patients with breast cancer [23], through the production of resolvins. During inflammatory responses, ω-3 PUFAs have been shown to help reduce levels of arachidonic acid (ARA), which is one of the main types of omega-6 PUFA (ω-6 PUFA) precursors to several potent pro-inflammatory mediators, including prostaglandins and leukotrienes [24]. Additionally, their ability to integrate into cell membranes inhibits enzymatic activities and interacts with signaling mediators, thereby reducing cell proliferation [25,26] or modulating the inflammatory process [27]. ω-3 PUFAs can potentiate the therapeutic action by changing the gene expression of specific genes, such as vascular endothelial growth factor (VEGF), which can affect the development of new blood vessels [28], and subsequently the occurrence of metastases [25,29], as well as the induction of the cell cycle arrest, apoptosis [25,30,31], and the production of lipid mediators [13,23,32,33,34].

Consequently, nutritional therapies can be considered a crucial component of the multimodal therapy approach in BC patients [3]. Diets containing omega-3 PUFAs have also been correlated with benefits against other common inflammatory diseases, and a reduced risk of several cancers, including colorectal, prostate, and BC [5,35,36,37,38].

The purpose of this study was to carefully review research published in the last 5 years on the use of ω-3 PUFAs as a preventive measure or treatment, including whether they were combined with antitumor drugs or not, and in animal models of BC. This review also sought to analyze the characteristics of the studies regarding the animal model employed, the therapeutic approaches (FAs and antitumor drugs) used, the primary components of ω-3 PUFAs, and the main study outcomes in relation to BC.

## 2. Methods

### 2.1. Search Strategy

This systematic review searched articles that were published in the last 5 years, including the years between June 2017 and September 2022, in the main databases, PubMed and Scopus. The indexed articles were selected, following the Preferred Reporting Items for Systematic Reviews and Meta-Analyses (PRISMA) guidelines [15]. The criteria of interest selected were keywords in the following sequence: ((Omega) AND (Breast tumor)), using the Boolean operators (DecS/MeSH):

SCOPUS: ((((TITLE-ABS-KEY (“omega-3”) OR TITLE-ABS-KEY (“n-3 fatty acids”) OR TITLE-ABS-KEY (“n-3 PUFA”) OR TITLE-ABS-KEY (“n-3 PUFA”) OR TITLE-ABS-KEY (“n-3 polyunsaturated fatty acids”) OR TITLE-ABS-KEY (“eicosapentaenoic acid”) OR TITLE-ABS-KEY (“docosahexaenoic acid”) OR TITLE-ABS-KEY (“ω-3 PUFAs”) OR TITLE-ABS-KEY (“LCn-3 PUFA”) OR TITLE-ABS-KEY (“long-chain n-3 polyunsaturated fatty acids”) OR TITLE-ABS-KEY (“n-3 FA”) OR TITLE-ABS-KEY (“ω-3 polyunsaturated fatty acids”) OR TITLE-ABS-KEY (“Omega-3-Acid”) OR TITLE-ABS-KEY (“Omega 3 Fatty Acids”) OR TITLE-ABS-KEY (“n3 Polyunsaturated Fatty Acid”))) AND ((TITLE-ABS-KEY (“breast cancer”) OR TITLE-ABS-KEY (“breast tumor”) OR TITLE-ABS-KEY (“Cancer of Breast”) OR TITLE-ABS-KEY (“Mammary Cancer”) OR TITLE-ABS-KEY (“Breast Neoplasms”)))) AND NOT (TITLE-ABS-KEY (“clinical trial”))) AND NOT (TITLE-ABS-KEY (meta-analysis)) AND (LIMIT-TO (PUBYEAR, 2022) OR LIMIT-TO (PUBYEAR, 2021) OR LIMIT-TO (PUBYEAR, 2020) OR LIMIT-TO (PUBYEAR, 2019) OR LIMIT-TO (PUBYEAR, 2018) OR LIMIT-TO (PUBYEAR, 2017)) AND (LIMIT-TO (DOCTYPE, “ar”)) AND (LIMIT-TO (LANGUAGE, “English”)).

PubMed: Search: (((((((((((“breast cancer” [Title/Abstract]) OR (“breast tumor” [Title/Abstract])) OR (“Cancer of Breast” [Title/Abstract])) OR (“Mammary Cancer” [Title/Abstract])) OR (“Breast Neoplasms” [Title/Abstract])) AND (((((((((((((((“omega-3” [Title/Abstract]) OR (“n-3 fatty acids” [Title/Abstract])) OR (“n-3 PUFA” [Title/Abstract])) OR (“n-3 PUFA” [Title/Abstract])) OR (“n-3 polyunsaturated fatty acids” [Title/Abstract])) OR (“eicosapentaenoic acid” [Title/Abstract])) OR (“docosahexaenoic acid” [Title/Abstract])) OR (“ω-3 PUFAs” [Title/Abstract])) OR (“LCn-3 PUFA” [Title/Abstract])) OR (“long-chain n-3 polyunsaturated fatty acids” [Title/Abstract])) OR (“n-3 FA” [Title/Abstract])) OR (“ω-3 polyunsaturated fatty acids” [Title/Abstract])) OR (“Omega-3-Acid” [Title/Abstract])) OR (“Omega 3 Fatty Acids” [Title/Abstract])) OR (“n3 Polyunsaturated Fatty Acid” [Title/Abstract]))) AND ((“2017/01/01” [Date–Publication]: “3000” [Date–Publication]))) NOT (review [Publication Type])) AND (english [Language])) NOT (“systematic review” [Publication Type])) NOT (Meta-Analysis [Publication Type])) NOT (“Clinical Trial” [Publication Type]) Filters: Other Animals.

### 2.2. Inclusion Criteria

The review included only original full-text articles written in English, published within the last 5 years, and that had used different inductions of BC in animal models (transgenic or induction by tumor cells or drugs) to evaluate the effect of dietary enrichment with ω-3 PUFAs or the use of ω-3 PUFAs in the treatment or prevention of BC, combined or not with other antitumor treatments. From the perspective of the patient, and the intervention, comparison, and outcome (PICO) criterion, the study problem was the unclear effect of ω-3 PUFAs on BC. Intervention: FAs supplementation before and after the induction of BC; Comparison: to assess the real benefits of the use of ω-3 PUFAs as a coadjuvant in the prevention or treatment of BC; Outcome: alternative treatment of BC.

### 2.3. Exclusion Criteria

We excluded studies based on the following criteria: (i) reviews, (ii) publications written in languages other than English, (iii) indexed articles published in more than one database (duplicates), (iv) only in vitro studies, (v) only clinical data analyses, and (vi) diets without ω-3 PUFAs.

### 2.4. Data Compilation

In this review, eleven of the authors (M.H.T., M.P.N., A.H.A., O.F.M.D., J.B.M., M.M.G., F.A.O., G.N.A.R., N.M.E.V., G.C., and L.F.G.), in pairs, independently and randomly analyzed, reviewed, and assessed the eligibility of titles and abstracts according to the strategy of established search. The authors M.H.T., M.P.N., A.H.A., O.F.M.D. and L.F.G. selected the final articles by evaluating the texts that met the selection criteria. The authors M.H.T., M.P.N., A.H.A., O.F.M.D., J.B.M., and L.F.G. were responsible for the search for the experimental design and characteristics of the diet and complementary treatment. The authors M.H.T., M.P.N., M.M.G., F.A.O., G.N.A.R., N.M.E.V., G.C., and L.F.G. searched for the characteristics of the BC cell induction and the experimental models. All authors contributed to writing the entire text of this review.

### 2.5. Data Extraction

The papers under evaluation were analyzed using five topics, which were represented in tables that addressed the following characteristics: (1) characteristics of breast tumor experimental models; (2) characteristics of the tumor cell transplantation model; (3) design of the experiment based on the FAs dietary supplementation with or without antitumor treatment; (4) main outcomes; (5) dietary supplements based on the types of omega-3-6-9 FAs utilized and their components.

### 2.6. Risk of Bias Assessment

The selection of articles was performed in pairs and a third independent author decided if the articles should be included. The data selected in the tables were divided by the authors into the groups already described above, and the checking of the data was carried out by the respective group. The final inclusion of studies into the systematic review was by agreement between all reviewers.

### 2.7. Data Analysis

To highlight the key traits, peculiarities, and exceptions, according to the applicability, the data gathered in each of the tables were evaluated in percentages and ranges of distribution. The frequency and quantity distribution of each type of the omega-3-6-9 FAs were also examined in the chosen studies. All data are considered to be the changes in the treated or experimental group (ω-3 PUFA, drug, or a combined treatment) compared to the control group (vehicle or FAs) data for each study.

## 3. Results

### 3.1. Overview of the Screening Process of the Included Studies

Following the inclusion and exclusion criteria described above, we found 679 papers in the last five years throughout the PubMed and Scopus databases, with 216 from PubMed and 463 from Scopus. Of the 216 articles identified from PubMed, 174 were excluded after screening because 112 studies were with humans, 39 were reviews, 18 were clinical trials, 2 were meta-analyses, 1 study was written in a language other than English, and 2 were editorials. After the eligibility assessment, a further 23 of the 42 studies were excluded, in which 13 showed in vitro studies only, 4 did not use a BC model for the ω-3 PUFAs’ dietary supplementation, and in 6 studies the diet did not include ω-3 PUFAs. Of the 463 articles found in Scopus, 450 were excluded after the screening, 183 studies were with humans, 155 were reviews, 27 were clinical trials, 14 were meta-analyses, 2 studies were in a language other than English, 13 were editorials, 14 were book chapters, 6 were communications, and 36 were duplicates of articles from the PubMed database. The eligibility analysis excluded a further five articles—three studies reported only in vitro results and two did not use the BC model. Thus, only 27 unduplicated full-text articles [17,18,19,39,40,41,42,43,44,45,46,47,48,49,50,51,52,53,54,55,56,57,58,59,60,61,62] were included in this systematic review, 19 from PubMed and 8 from Scopus, as shown in Figure 1.

### 3.2. Characteristics of the Breast Cancer Experimental Animal Model

The influence of an omega-3-6-9 FAs-enriched diet, combined or not with another antitumor treatment, on the prevention and treatment of BC was evaluated in this systematic review; the evaluation specifically focused on the omega FAs, combined or not with another antitumor treatment, evaluating their influence as the coadjuvant in conventional tumor treatment. The last five years has shown increasing attention to the real benefits of ω-3 PUFAs and their types and a more in-depth search of their actions against BC. In this first analysis, we describe the characteristics of BC induction in animal models used in the selected studies, as shown in Table 1, detailing the type of model, source of breast tumor, and the animal characteristics.

The breast tumor experimental model most used in the selected studies was the conventional tumor cell transplant in specific tissue regions of animal models (69%) [17,18,19,39,42,43,45,46,49,50,51,53,55,57,58,59,60,61,62], followed by the model of spontaneous tumor generation using transgenic animals (17%) [40,41,47,52,55,56], and lastly, 14% of studies induced BC in the animal model by carcinogenic drug transplant [44,48,54].

All studies utilized the in vivo murine model—93% mice [17,18,19,39,40,41,42,43,44,45,46,47,49,50,51,52,53,55,56,57,58,59,60,61,62] and 7% in rats [48,54]. The breast tumor model by cell transplant was performed in 64% of studies using different cell sources, 70% from humans [17,18,19,45,49,50,51,53,55,58,59,60,61,62] and 30% from mice [39,42,43,46,53,57], and in 17% of studies, transgenic animals developed the tumor spontaneously [40,41,47,55,56]. Chemical induction of the tumor occurred in 15% of studies—75% by the 7,12-dimetilbenz(a) antraceno (DMBA) drug [44,52,54] and 25% by the n-methyl-n-nitrosourea (NMU) drug [48]. Curiously, all rats were Sprague–Dawley (wildtype) and the tumors in these animals were induced by drugs [48,54].

Regarding the mice strain, the studies used mainly Balb/c (57%) [18,19,39,42,43,44,45,51,53,55,57,58,59,60,61,62], and among this group 44% were the nude type (*nu*/*nu*) [18,19,45,51,55,57,58,59,60,61,62], 44% were the wildtype [39,42,43,44,53,57,60], and two mice showed genetic modifications (*Foxn1nu* [58] and *J:nu* [19]). The breast tumor model by cell transplant used Balb/c mice almost exclusively, and in only one study the Balb/c wildtype had the tumor induced by a drug [44]. The NOD SCID Gamma (NSG) mice were used in 14% of the selected studies [17,49,50,53] for the tumor model by cell transplant—75% of these mice showed genetic modifications (NOD.Cg-*Prkdc^scid^Il2rg^tm1Wjl^*/SzJ) [17,49,50] and 25% were wildtypes [53]. The study by Liu [46] used C57BL/6 mice and compared two different genotypes of adipose fatty acid binding protein-deficient mice (A-FABP^−/−^) with the wildtype, and the study by Li [52] used the offspring of the wild-genotype mice from Fat-1 and C57BL/6 crossing for the tumoral chemical induction model, totaling 11% of the selected studies [42,46,52]. The remaining 18% of the studies used transgenic mice [40,41,47,55,56], of which 80% were obtained from crossing the FVB/N mice with the mammary tumor virus (MMTV), so that these mice had the MMTV-*Neu^ndl^*YD5 genotype [40,47,56], with only one having the FVB/N-Tg(MMTV-PyVT)634Mul genotype [55]; 20% were obtained from crossing the SV129 mice with the c(3)1-Tag, using the hemizygous pups’ genotype [41].

Female murines were predominantly used in the selected studies (83%) [17,18,39,40,41,42,44,45,47,48,49,50,51,52,53,54,55,56,57,58,59,61,62], with male-only in 3% [19], both animal sexes in 7% [43,60], and in 7% of studies this information was not reported [46]. Regarding the animals’ ages, 36% of the selected studies used young mice in the development phase, from 3 to 5 weeks [18,40,41,49,52,55,56,58,59,61], 36% of studies used adult mice at 6 weeks of age [17,39,45,50,51,53,55,57,62], 14% used adult mice between 7 and 8 weeks of age [39,42,44,53], and 14% did not report age [19,43,46,47,60].

### 3.3. Breast Cancer Cell Transplant Models

The conventional model for inducing breast tumors by cell transplant was the most used in the selected studies (69%) [17,18,19,39,42,43,45,46,49,50,51,53,55,57,58,59,60,61,62]; the main tumor tissue source was mammary adenocarcinoma in 89% of studies [18,19,39,42,43,45,46,49,51,53,55,57,58,59,60,61,62], followed by the adenocarcinoma (MAXF401 cells) and the invasive ductal carcinoma (MAXF574 cells), both used with 9% of frequency among the selected studies [17,50], as shown in Table 2. This last tumor cell type has similar tissue, but its characteristics are more aggressive, invasive, and metastatic. A varied type of cell lineage induced the tumor in the mammary adenocarcinoma tumor model; the most frequently used were the MDA-MB-231 cells (35%) [45,51,55,58,59,61,62], followed by the 4T1 cells (17%) [39,42,53,57], the MCF7 cells (9%) [18,19], and, in a lower incidence (4%), the LMM3, EO771, MMT060562, BCX010, and SUM149PT cells [43,46,49,60]. According to the classification of BC tumor subtypes, we identified that most (80%) of the selected studies used some of TNBC tumor types such as the MDA-MB-231, MAXF401, MAXF574, SUM149PT, 4T1, BCX010, LMM3, and MMT060562 cells [17,39,42,43,45,46,49,50,51,53,55,57,58,59,61,62], and already in some studies there was evidence of the use of Luminal A BC types (8% of MCF-7 cells) [18,19], and in 4% of studies the Luminal B BC-type cells (E0771 cells) were used [46]. The LM3 cells [60] did not show a clear classification of the BC type.

The culture medium most used in the support of the tumor cells’ growth was Gibco Dulbecco’s Modified Eagle Medium (DMEM) (53%) [18,19,49,53,55,57,58,59,61,62], mainly supplemented with 10% of fetal bovine serum (FBS). However, 16% of the selected studies did not report the type of medium used [17,39,50] and other studies used similar culture mediums with a lower quantity of nutrition, such as a Modified Eagle Medium (MEM) (11%) [43,60], or with a higher concentration of glucose, such as the Roswell Park Memorial Institute (RPMI) (11%) [42,46] or Iscove’s Modified Dulbecco (MD) medium (11%) [45,51]; these also had a higher concentration of sodium pyruvate, additional amino acids, a (4-(2-hydroxyethyl)-1-piperazineethanesulfonic acid (HEPES) buffer, selenium, and other components.

Only mice were used for the cell implantation breast tumor model, mainly immunodeficient mice such as Balb/c nude (43%) [18,19,45,51,55,59,61,62], Balb/c wildtype (29%) [39,42,43,44,48,53,57,60], and NSG (19%) [17,49,50,53], as well as the C57BL/6 mice used in 14% of studies [46,52]. These mice also showed some genetic modifications: 75% were NSG mice [17,40,50], 50% C57BL/6 [46], 22% Balb/c nude [19], and 17% of Balb/c wild-type mice. The number of cells implanted and the volume administered during the tumor induction varied according to the local cell implantation. In specific regions such as the mammary gland, including the inguinal mammary fat pad, the number of cells was from 5 × 10^3^ to 5 × 10^5^, using about 50 to 100 µL; in the wide-body regions such as the flank, armpit, hind thigh, and scapula the number of cells was from 10^6^ to 1.5 × 10^7^, using about 80 to 200 µL of volume. However, it was noted that in the study by Newell [17], the tumor induction model was carried out by transplanting a portion of the tumor tissue sectioned into 30mm^3^ slices rather than by injecting tumor cells.

The vehicle used during tumor induction was not reported in 40% of selected studies [18,39,43,46,49,50,55,59], and among the studies that did report this, 21% used Phosphate-Buffered Saline (PBS) with Matrigel [53,58,61,62], 10% used Iscove’s MD medium and PBS [19,45,51,53], and less frequently (5%), the culture medium itself or other solutions [42,57,60,62] were used. The breast tumor induction model by cell implantation in 65% of the studies was xenograft [17,18,19,45,49,50,51,53,55,58,59,61,62] and in 35% it was allograft [39,42,43,46,53,57,60].

### 3.4. Experimental Design and Characteristics of the Diet and Complementary Treatment

Table 3 analyzes the in vivo study experimental design and the characteristics of the groups regarding the omega-3-6-9 FAs use and antitumor treatment on the BC model. The number of groups used by the studies varied from two to nine, with a minimum sample size of four and a maximum of 20 animals per group, according to the number of comparisons used. The study by Abbas [44] that used 50 and 100 animals per group in different stages of the study was an exception. The animal diets were administered ad libitum in all studies, using, mainly, three types of diet: 32.1% used a standard diet (one case with free selenium and another supplement) [17,42,47,48,53,54,55,60,61]; 14.3% used the AIN-76 or AIN-93 diet, or with modifications (AIN-76A or AIN-93G) [41,43,44,45,49,51,52,56]; and 14.8% used one of five diets—the Chow diet, the LabofeedH diet, the Lieber–DeCarli diet, an experimental diet, or a diet with different fat percentages [19,40,46,57]; and 21.4% did not provide any references regarding the peculiarities of the diet [18,39,50,58,59,62].

The proposal for the use of ω-3 PUFAs on BC models in the selected studies was 14.8% for the prevention of tumor incidence [43,44,52,57], 18.5% for prevention and antitumor treatment [19,41,47,54,56], and 11.1% for treatment-only [40,46,60]. However, most of the studies analyzed ω-3 PUFAs combined with antitumor drugs, aiming for the prevention and treatment of BC in 18.5% of studies [17,48,50,51,61], or for treatment-only in 37% of studies [18,39,42,45,49,53,55,58,59,62].

All studies used at least one ω-3 PUFA source, mainly associated or not with ω-6 PUFAs and ω-9 MUFAs, detailed in Table 3, as shown by the total amount of omega-3-6-9 FAs used in the experimental groups of each study; however, this information was reported in only 59.3% of studies on ω-3 PUFAs [17,19,40,41,42,43,45,46,47,48,49,50,51,54,55,56,59,60], 37% on ω-6 PUFAs [17,41,43,45,47,50,51,54,56,60], and 33.3% on omega-9 monounsaturated fatty acids (ω-9 MUFAs) [17,43,45,47,50,51,54,56,60]. Some studies reported the amount used by the types of omega-3-6-9 FAs, as explored below.

The main sources of ω-3 PUFAs were fish oil (29.6%) [41,46,48,49,56,57,58,61], menhaden oil (11.1%) [40,46], chia and flaxseed oils (7.4% each) [43,60], and safflower oil (7.4%) [54,56], followed by the DHAsco diets [50,51], which is a specific diet rich in the long-chain polyunsaturated fatty acids (LCPUFAs), DHA, from marine microalgae *Crypthecodinium cohnii* (14.8%), and the use of pure ω-3 PUFAs, such as DHA (22.2%) [18,39,42,46,48,60] and EPA (11.1%) [42,46,48,49,55]. The main ω-6 PUFA sources were corn oil (33.3%) [19,40,41,43,44,57,58,60,61,62], safflower oil (18.5%) [43,47,51,54,56,57], and canola oil (14.8%) [41,44,51], followed by the oils of lard, olive, and soybean (11.1% each) [17,18,40,46,47,51,52,57], the ARAsco diet (7.4%) [50,51]—a specific diet rich in the LCPUFAs ARA from the soil fungus *Mortierella alpina*, flaxseed oil (7.4%) [47,56], and the use of linoleic acid (LA) ω-6 PUFA (7.4%) [45,54]. The use of vegetable oil, peanut oil, rapeseed oil, sunflower oil, and cocoa butter was reported in low frequency (3.7%) [39]. The ω-9 MUFA is part of the composition of almost all oils described above and its use was quantified in only 33.3% of the studies [17,43,45,47,50,51,54,56,60].

The FAs administration was mostly oral via in-food supplementation (70.4%) [17,40,41,43,44,45,46,47,48,49,50,51,52,55,56,57,58,60,61]; three studies used gavage (11.1%) [19,42,54], 7.4% used more specific means via intravenous and intraperitoneal administration, and the study by Zhu [59] did not report on this (3.7%). The timeframe for the FAs treatment was from 3 weeks to 5 months in 44% of the studies that evaluated only ω-3 PUFAs as a prevention or treatment measure for cancer [19,40,41,43,44,46,47,52,54,56,57,60]. In 30% of the studies that used ω-3 PUFAs and an antitumor drug at the same time [18,39,42,45,49,53,55,58,62], the duration of this combined treatment varied from a single dose to 6 weeks, and in 48.1% of these studies, the FAs were administrated from 1 to 3 weeks before the antitumor drug treatment, and continued in alignment with the timing of the drug administration [17,19,41,43,47,48,50,51,52,54,55,57,61]. Of the antitumor drugs used in 56% of studies [17,18,39,42,45,48,49,50,51,53,55,58,59,61,62], 20% was Docetaxel (DTX) [17,39,48,51], 13.3% was Doxorubicin (DOX) [45,50], and 6.7% were others such as Taxol (TAX), Adriamycin (Adr), Avastin (Ava), Dasatinib, Phloridzin (PZ), Rapamycin, Sorafenib (SFN), Retinoic acid, Disulfiram (DSF), and 12-(3adamantan-1-yl-ureido)-Dodecanoic Acid (AUDA) [18,42,49,53,55,58,59,61,62]. Only three studies used ω-3 PUFAs as the nanoparticle stabilizer to improve the antitumor drug efficacy [18,39,53]; in these cases, the animal model received a regular diet without FAs enrichment.

Figure 2 shows the main results of the studies included, presenting the effects of ω-3 PUFAs associated, or not, with the antitumor drug treatment in different aspects of tumor analysis. The spider chart and purple-bar graphic of Figure 2A–I shows that 92.6% of the studies analyzed the tumor growth by volume [17,18,39,40,42,43,44,45,46,47,48,49,50,51,52,53,55,56,57,58,59,60,61,62] (Figure 2F), 85.2% through molecular and histological analyses [17,39,40,41,42,43,44,45,46,48,49,50,51,52,53,55,57,58,59,60,61,62] (Figure 2I), 77.8% analyzed the animal weight [17,18,19,39,40,41,42,45,46,48,49,50,51,52,54,55,56,57,58,59,62] (Figure 2D), 70.4% used tumor weight and genetic assessment [17,19,39,40,42,45,46,47,50,52,53,54,55,56,57,58,59,60] (Figure 2E,I), and less often the analysis was through tumor metastasis (33.3%) [42,43,47,52,53,54,56,57,60] (Figure 2G), tumor incidence (25%) [41,44,47,52,54,56,57,60] (Figure 2B), tumor latency (Figure 2C), and animal survival (22.2%) (Figure 2H) [47,52,54,56,60]. The box plots around the spider chart represent the analysis of the results for each specific outcome comparing the difference (percentage or days) between the results of the experimental group versus their control group of each study and in each condition (FAs only are in green, drugs only are in red, and both treatments combined are in orange).

The tumor incidence was analyzed only by the studies that used different types of omega-3-6-9 FAs in the enriched diet as antitumor treatments, comparing the predominance of ω-3 PUFAs with other FA types [47,52,54,56,60]. The mean incidence delay was 20%, when considering 50% of animals having been affected by tumors (ranging from 9 to 27%) when using a diet enriched with ω-3 PUFAs compared to one enriched with ω-6 PUFAs. However, when considering 100% of animals having been affected by tumors, the mean incidence delay was 12% (ranging from 0 to 23%), in the same comparison based on the types of FAs (the amount of ω-3 PUFAs was higher than ω-6 PUFAs). The tumor latency was also analyzed in studies that only compared the different types of FAs (e.g., the amount of ω-3 PUFAs was higher than ω-6 PUFAs), performed by T50 and T100 (the interval from when cancer started until it was diagnosed in 50% and 100% of animals, respectively) [40,44,47,52,54,56,57,60]. The T50 mean was 19 days of delay and the minimum and maximum delay times were 7 and 50 days, respectively; already in latency in T100, the mean delay was 13 days, and the minimum and maximum delay times were 5 and 18 days, respectively.

We analyzed the percentage of animal weight change in the selected studies according to the results on the effect of using an ω-3 PUFA-enriched diet as an antitumor treatment, or only an antitumor drug treatment, and the combined treatment (ω-3 PUFAs plus an antitumor drug). A diet enriched more with ω-3 PUFAs than ω-6 PUFAs showed a mean reduction of 46% in animal weight, varying from 1% to 151%, in which the last was a 51% weight gain (median 18%). In the experimental groups that analyzed only the drug effect, when compared with the control group (vehicle), the majority of animals showed a weight gain up to 10%, and in the combined treatment versus the control group, animal weight increased in the majority of studies (median 103%, with 3% of gain). There was a reduction between 2 and 30% (mean 73%) in only a few cases. The tumor weight was also analyzed under the three different experimental conditions: the combined ω-3 PUFAs and antitumor drug treatments versus the control was more effective, reaching 86% of tumor weight reduction (mean 57%), with the use of only an FAs-enriched diet (the amount of ω-3 PUFAs was higher than ω-6 PUFAs) the mean was 43%, and with the drugs-only versus the control, the tumor weight reduction was 32%.

The tumor growth was reported at different times of tumor evolution; therefore, we grouped the results in 15-day periods for this analysis and considered the mean results of each study for each condition. In the early stage, the highest reduction of tumor growth (a mean of 77% and a maximum value of 87%) occurred with the combined ω-3 PUFAs and drug treatment versus the control group (until 15 days), from 30 to 45 days the mean reduction stayed the same (53%), but in the late stage the mean reduction for this comparison decreased to 47%, being more effective in the groups that used the ω-3 PUFA-enriched diet only, compared with a diet enriched with a higher amount of ω-3 PUFAs than ω-6 PUFAs (mean 49% and maximum value 79%). Thus, the ω-3 PUFA-enriched diet was more effective in studies that followed the animal for a longer period (140 and 300 days). In the experimental groups that analyzed only the antitumor drug effect versus the control, the tumor size reduction was less effective (mean of 17%, 24%, and 7% until 15, 30, and 45 days, respectively).

The tumor metastasis was analyzed mainly in the studies that used only a ω-3 PUFA-enriched diet as the tumor treatment, showing a mean reduction of 37%, varying from 14% to 70% when the diet was predominantly ω-3 PUFAs versus ω-6 PUFAs; however, the combined treatments (ω-3 PUFA plus drug) versus the control was more effective, reducing the metastasis mean to 70%, ranging from 61 to 84%. In the groups that used drugs only versus the control, the results were less effective with a reduction of 17% and 60%, and an increase of 26% (in the graphic of Figure 2 this shows as 126%). The survival analysis of the animals in the studies is reported by the Kaplan–Meier curves at different times; the common time of most studies was 50% (T50 of box plot of Figure 2), and only one study reported the analysis of 80% onwards (T80). These percentages (T50 or T80; 50% or 80%) represent the fraction of animals surviving divided by the number of animals at risk for a certain amount of time after treatment. In the studies that used the FAs-only comparison, the increased survival was 20 days (T50) and 7 days (T80) when the diet was predominantly ω-3 PUFAs versus ω-6 PUFAs. For the combined treatment the survival increase was more remarkable with a mean of 23 days ranging from 5 to 45 days, compared to their respective control groups.

The molecular analysis was mainly evaluated by the growth pathways’ cell proliferation and survival, followed by an inflammatory response, cell death pathways, energy and metabolism homeostasis, immune response mediators, cell cycles, synthesis of FAs, immunophenotype characterizations, necroptosis, checkpoint immunology, and the mechanism of deoxyribonucleic acid (DNA) repair. Genetic evaluation was reported in 70.4% of the selected studies [17,19,40,41,43,44,46,49,51,52,53,55,56,57], focusing on the gene up or down-regulation from the diet intervention or drug treatment, or both interventions combined, using different assays according to the proposal of each study. The study by Newell [51] was the most in-depth genetic screening. The histological analysis was a complimentary evaluation that focused mainly on apoptosis detection used in 39% of studies [39,40,42,43,44,46,49,50,51,52,55,57,59,60,61,62], followed by cell proliferation in 22% [41,43,50,52,55,59,60,61], tissue necrosis in 14.6% [17,45,48,51,52,62], and, less often, metastasis (9.8%) [42,53,57,60], angiogenesis (7.3%) [48,53,57], autophagy (4.9%) [55,59], and infiltration/invasion (2.4%) [43].

### 3.5. Dietary Compounds

Analyzing the main types of each PUFA (ω-3 and ω-6) and ω-9 MUFA, we observed that some types were more frequently cited than others and with this detailed analysis, we tried to verify if there was a trend for the most representative or important types of FAs, as shown in Table 4 and Figure 3A-D.

The three most important types of ω-3 PUFAs (ALA, EPA, and DHA) were the most frequently used in the selected studies—63% [17,42,43,44,45,46,48,49,51,52,54,55,57,61,62] used ALA or DHA and 44.4% used EPA [19,40,41,42,46,47,48,49,55,56,58,59]; however, other ω-3 PUFA types were also used, such as stearidonic acid (SDA) in 11.1% of studies [17,47,56] and eicosatetraenoic acid (ETA) or docosapentaenoic acid (DPA) [47,56] in 7.4% of studies, as shown in the green spider chart of Figure 3. Almost all studies reported the amount of each type of ω-3 PUFA; only 29.6% of the studies did not mention the quantity of ω-3 PUFA type used [18,44,52,53,57,58,61,62], and in two studies (7.4%) there was no mention of the quantity used and ω-3 PUFAs were not directly applied in the diet [18,39].

The type of ω-6 PUFA most used in the dietary supplementation of the selected studies was LA in 70.4% of studies [17,19,40,41,43,44,45,47,48,50,51,52,54,56,57,58,60,61,62], followed by 25.9% of studies using Arachidonic acid (ARA) [17,19,43,45,47,51,56], 18.5% using Gamma-linolenic acid (GLA) [45,47,50,51,56], 14.8% using Eicosadienoic acid (EDA) [41,47,50,56], and 7.4% using Dihomo-gamma-linolenic acid (DGLA) or Docosadienoic acid (DDA) or Adrenic acid (AdA) [47,56], as shown in the pink spider chart of Figure 3. For 63% of studies the amount of each type of ω-6 PUFA used was not reported [18,19,39,40,42,44,46,48,49,52,53,55,57,58,59,61,62], and 25.9% of studies did not use any ω-6 PUFA type [18,39,42,49,53,55,59].

In addition to being the most important type of ω-9 MUFA, oleic acid (OA) was the most frequently used in 63% of studies [17,40,43,44,45,47,48,50,51,52,54,56,57,58,60,61,62], followed by erucic acid (EA) or nervonic acid (NA) in 7.4% of studies [47,54,56], as shown in the blue spider chart of Figure 3. Only 33.3% of studies reported the amount of the ω-9 MUFA type used [17,43,45,47,50,51,54,60], and 25.9% did not use any ω-9 MUFA in the dietary supplementation [18,39,42,49,53,55,59].

Of the selected studies, 26% used only pure ω-3 PUFAs in the study [18,39,42,49,53,55,59], but the majority of studies used different sources of FAs that also comprised ω-6 PUFAs and ω-9 MUFAs, in different proportions, allowing the comparison between a diet enriched with sources of, predominantly, ω-3 PUFAs versus ω-6 PUFAs.

## 4. Discussion

This systematic review showed that the effects of ω-3 PUFA supplementation was predominantly (55.6%) [17,18,39,42,45,48,49,50,51,53,55,58,59,61,62] analyzed combined with some type of conventional antitumor treatment applied in different types of BC-induced models. Less frequently (44.4%) [19,40,41,43,44,46,47,52,54,56,57,60], although with relevance, the studies analyzed the effect of enriching the diet with ω-3 PUFAs, without combining them other antitumoral treatments, as prevention and/or treatment for BC in relation to another PUFA supplement, applied mainly in transgenic BC models [40,41,47,56] and BC chemically induced models [44,52,54].

A current review showed that ω-3 PUFA supplements combined with chemotherapy and radiotherapy for BC patients can reduce the pain symptoms, prevent cachexia-anorexia syndrome, increase the weight of cancer patients, decrease the process of mitosis and cancer cell proliferation, as well as reduce the inflammatory response and support chemotherapy treatment, and improve the cancer patient’s overall survival rate [63]. Moreover, according to a clinical review, women with high intakes of total ω-3 PUFAs in comparison to ω-6 PUFAs have been found to have a lower risk of BC [24]. In addition, by analyzing eating habits, a clinical trial showed a good association between total MUFAs, OA, and palmitic acid (C16:0) and an increase in BC risk [64]. Nonetheless, the effects of ω-3 PUFAs on tumor treatment are controversial [65]. Some studies suggest that ω-3 PUFAs may prevent or slow cancer development by reducing inflammation, slowing cancer cell growth and division, and preventing the formation of new blood vessels [66].

Due to the significant differences in the experimental model between studies, the ω-3 PUFA source (fish oil, menhaden oil, and chia oil), ω-3 PUFA type (ALA, EPA, and DHA), total dietary amount administered, route of administration (diet or gavage), and time of use, as well as accounting for the other FAs, it was difficult to determine the appropriate dose of ω-3 PUFAs for BC prevention or treatment in the selected studies. However, according to the global standard for EPA and DHA intake, the dose of supplemental ω-3 PUFAs known to be beneficial for some treatments or aging processes for humans is already well-established, such as cardiovascular diseases (at least 500 mg/day of EPA+DHA), pregnant/lactating women (D–A—300 mg/day), and general adults (300–400 mg EPA+DHA/day). In Brazil, the standard was reported with a minor alteration, for coronary artery disease (1 g/day EPA+DHA) and pregnant/lactating women (D–A—200 mg/day). The recommended daily dose in France for reducing the risk of breast and colon cancer was 500 mg of EPA+DHA [67]. The dose that was advised in 2017 for cardiovascular diseases (greater than 1 g/day) and pregnant/lactating women (D–A—700 mg/day) was slightly raised [68]. A current review study showed that the dose of DHA of 2 g per day shows evidence for safety, good absorption, and saturating plasma levels according to distribution, interconversion, and dose response in humans [69].

ALA, DHA, and EPA were the main ω-3 PUFAs (44 to 63%) reported by the selected studies, and their antitumoral effect was analyzed as either combined with antitumor drugs or compared with a diet with a higher intake of the LA ω-6 PUFA type (70.4%), or the ARA and GLA ω-6 PUFAs (about 20%) and the OA ω-9 MUFA (63%). EPA and DHA carried by apo B-containing lipoproteins (Non-High-Density Lipoprotein (HDL)) showed a protective effect on tumor cell proliferation only in tumors negative for ER and PgR, comparing different lipoproteins and the degree of severity of BC in women [70]. Furthermore, in a Swedish cohort study, postmenopausal women who consumed more ω-6 PUFAs and fewer heterocyclic amines had an increased risk of developing BC [71].

The age of the animals was another crucial factor in how ω-3 PUFA-enriched diets affect BC incidence; younger animals showed a stronger protective effect than older animals [72]. Puppies as young as 3 weeks old were employed in the evaluation of the ω-3 PUFA effects in four of the chosen studies [41,44,52,56].

### 4.1. Outcomes of the Omega-3 PUFA Effect Combined with Antitumor Drugs or Not

There was a consistent tendency in the analyses of the studies’ results, although we found great heterogeneity in the experimental design between the selected studies. For example, the majority used dietary or antitumor drug intervention following tumor induction and detection [17,18,19,39,40,42,43,45,46,47,48,49,50,51,53,54,55,57,58,59,60,61,62]; some studies used maternal supplementation, followed by analyses of the offspring [41,44], while others used dietary intervention during the first few weeks of the animal’s life [44,52,56].

**The tumor growth and tumor weight** were the measures mostly used in the studies about disease progress, with 92.6% and 70.4% of studies reflecting this, respectively. For both measures, the analyses showed that ω-3 PUFAs combined with antitumor drugs were more effective in decreasing tumor weight and size in comparison to the controls, mainly in the early stage of tumor growth (until 15 days), but the continuous use of ω-3 PUFAs over time showed a higher reduction in tumor growth (mean of 11, 36, 46, and 49% until 15, 30, 45, and more than 140 days, respectively). The only group that used a longer timeframe for the analyses (140 and 300 days) showed that this was more effective than the combined treatment. The use of antitumor drugs only was not as effective.

**Weight loss** is frequently the first visible sign of cancer and may be easily controlled in both clinical and preclinical studies. However, as the disease progresses, cachexia can develop, which can aggravate the condition and increase mortality. This parameter was reported in 77.8% of studies; the antitumor drug groups in comparison to the control groups were more effective in avoiding weight loss and allowing weight gain than the combined treatment groups (ω-3 PUFAs and drug) or the FAs groups, in relation to their controls. All conditions had a weight gain percentage, highlighting the study of Monk [40] and Liu [46], which used obese animals in the FAs groups and showed the highest gain; however, this group also had a smaller mean reduction (46%) in comparison to the drug and combined treatment groups.

The metastasis, the incidence and latency of the tumor, and the survival rate, were analyzed in some studies (33.3, 25, and 22.2%, respectively) on the ω-3 PUFA effect, normally without antitumor drugs and mainly in the transgenic BC model and in BC chemically induced models. A higher **metastasis** reduction occurred during the use of ω-3 PUFAs combined with antitumor drugs (61 to 84%), followed by using only ω-3 PUFAs giving a 14% to 70% reduction. The effect of the antitumor drug compared to the vehicle (control) showed controversial results between different antitumor drugs, and either a decrease in metastasis of 17 and 60%, or an increase of 26% [42]. The role of ω-3 PUFA supplementation in the prevention of cancer metastasis was demonstrated in clinical studies in BC patients using ω-3 PUFAs in conjunction with antitumor drugs and mastectomy, and showed decreases in the presence of the proliferative marker Ki67 and the angiogenesis marker VEGF [73]. In advanced pancreatic adenocarcinoma patients there were benefits in terms of quality of life, reducing the metastatic burden, and a significant reduction in serum levels of pro-angiogenic markers [73,74]. A higher reduction in **tumor incidence** occurred when the ω-3 PUFA supplementation was higher than the ω-6 PUFAs without any antitumor drugs [41,44,47,52,54,56,57,60], with a mean of a 20% incidence of reduction in the evaluation of half of the tumor-free animals (50%), and a 12% reduction considering all animals analyzed (100%). The distribution of **latency** delay was similar for T50 and T100, with a median of 15 days; however, the follow-up of this analysis had a wide range from 20 to 300 days. Clinical studies reported the prevention effect of ω-3 PUFA intake in cancer and cardiovascular disease in adults more than 50 years old [24,64,75], and decreased latency was reported in preclinical studies [24].

In general, survival analysis aims to determine the proportion of a population that will survive through to a specific point in time. Different percentual times were used to evaluate the Kaplan–Meier, with the most common time in the studies being 50% (T50 of the box plot of Figure 2), and in only one study 80% and onwards (T80) was reported. A higher ω-3 PUFA-enriched diet (more ω-3 PUFAs than ω-6 PUFAs) showed a survival increase of 20 days, while the combined treatments (ω-3 PUFAs plus drugs) showed a mean increase of 23 days (ranging from 5 to 45 days of increase, as shown in the orange boxplot of Figure 2). Patients with locally advanced BC had dramatically extended overall survival and progression-free survival when combining the usage of an ω-3 PUFAs-enriched diet with chemotherapy and mastectomy [23].

**Molecular evaluation** was used for understanding the pathophysiology of BC in the selected studies through the protein expression analysis, investigating the inhibitory potential of ω-3 PUFAs in oncogenic pathways such as nuclear factor-kappa B, phosphatidylinositol-3-kinase/Protein kinase B/Mammalian target of rapamycin, and Wnt/β-catenin, as well as in the cell cycles (Cyclin B1, Wee1, and Poly (ADP-ribose) polymerase), energy homeostasis/metabolism (Glyceraldehyde–3–phosphate dehydrogenase, estrogen, insulin, resistin, and leptin), fatty acid synthesis (SREBP), oxidative stress (reactive oxygen species and hypoxia-inducible factors), DNA repair mechanisms (p53), and integrity and cell death through the processes of necrosis, apoptosis, and necroptosis (caspases, cytochrome C, tumor necrosis factor, receptor-interacting protein kinase, and mixed lineage kinase domain-like pseudokinase). The immunological analyses were focused on the immunophenotype characterization of tumors (through CD-type markers), the analysis of immune and inflammatory response mediators’ (interleukins and tumor necrosis factor-a) interaction capacity, and the activation of immune checkpoint receptors (Programmed death-1 and Programmed death-ligand 1). Furthermore, investigations into angiogenesis processes (signaling VEGF) have been observed, which have a significant impact on the occurrence of metastases [76]. Patients treated with chemotherapy and mastectomy for locally advanced BC, and supplemented with ω-3 PUFAs, showed significantly decreased expression levels of Ki-67 and VEGF, compared to the placebo [23].

In addition to molecular analysis, **histological analysis** was also carried out using traditional staining methods to evaluate the effects of ω-3 PUFAs on a variety of processes around BC progression, focusing primarily on apoptosis detection [39,40,42,43,44,46,49,50,52,55,57,59,60,61,62], cell proliferation [41,43,50,52,55,59,60,61], tissue necrosis [17,45,48,50,52,62], and less frequently, on tumor metastasis [17,45,48,50,52,62], angiogenesis [48,53,57], autophagy [55,59], and tumor cell infiltration and invasion [43]. In accordance with each study’s proposal, some of these carried out **genetic evaluation**, concentrating on the impact of ω-3 PUFA supplementation relative to, or not, antitumor medications in the gene up- or down-regulation.

### 4.2. Animal Models Used in the BC Induction

The **human tumor cell transplantation model** (xenograft transplantation) was the most common BC model used (67%), mainly through the MDA-MB-231 cell transplantation in immunodeficient mice. These TNBC cells have more aggressive growth and an intermediate response to chemotherapy [77]. This model is more advantageous in having short cycles, low costs, a minimal variation, and high rates of tumor growth in comparison to the induced models (biological, physical, and chemical) and the genetically engineered mouse models [78]. The local cell implantation ranged between studies, in orthotopic transplantation, such as in the mammary gland and the inguinal mammary fat pad, the number and volume of cells implanted were smaller than in ectopic transplantations such as in the flank and armpit, among others. An intraductal method of transplantation is thought to be preferable to the mammary fat-pad transplantation for orthotopic transplantation, due to the production of a better pathological microenvironment for BC cells; however, this method is technically more difficult and only allows for the injection of a limited number of cancer cells [78].

The **biological induction model of BC** mainly relies on lentiviruses to overexpress oncogenes or silence tumor suppressor genes in experimental animals. This model was reported in some studies (19%) using MMTV-Neundl and MMTV-PyVT. The replication-competent ALV–LTR splice acceptor (RCAS), the Avian Leukosis Virus (ALV) and the Long terminal repeat (LTR), which contains genes of interest, can be directly injected into the glands of MMTV transgenic mice to induce breast tumors. These mice develop tumors within three weeks, induced by RCAS viral particles encoding the T-antigen in the mouse Mammary Tumor virus-polyomavirus (PyMT) gene in mammary ducts. Other oncogenes such as Neu and Wnt-1 take about seven months [78]. **The chemically induced BC model using DMBA** [44,52,54], or MNU drugs [48], was also reported in some studies (15%). The animals that received DMBA administration showed PIK3CA gene mutations detected in all tumors, positively correlating with the activation of protein kinase B (AKT), and these genetic changes in the induced mammary tumors were similar to those that exist in natural mammary cancers [78].

The present study’s limitation was the wide range of experimental designs used in the studies that were selected, which made it difficult to make a more accurate comparison to determine the optimal dose of ω-3 PUFAs for the prevention of BC over the short and long-term and in conjunction with conventional antitumor therapies.

## 5. Conclusions

Omega-3 PUFA supplementation is an important coadjuvant to chemotherapy or other traditional antitumor therapies and shows remarkable results in combination with these other treatments, reducing tumor growth and weight (during the first fifteen days after tumor induction) compared to the isolated use of drugs or ω-3 PUFAs alone. Furthermore, the survival rate is increased. The effect of isolated ω-3 PUFAs was mainly analyzed in terms of tumor incidence and latency, as well as tumor metastasis in the transgenic BC model and chemically induced BC model, showing significant results in terms of animal weight (improving weight gain and preventing weight loss) and tumor growth reduction after prolonged use. However, the optimal ω-3 PUFA dose for pharmaconutrition and antitumor effects or the prevention of BC is unclear from the preclinical research.

## Figures and Tables

**Figure 1 nutrients-15-01310-f001:**
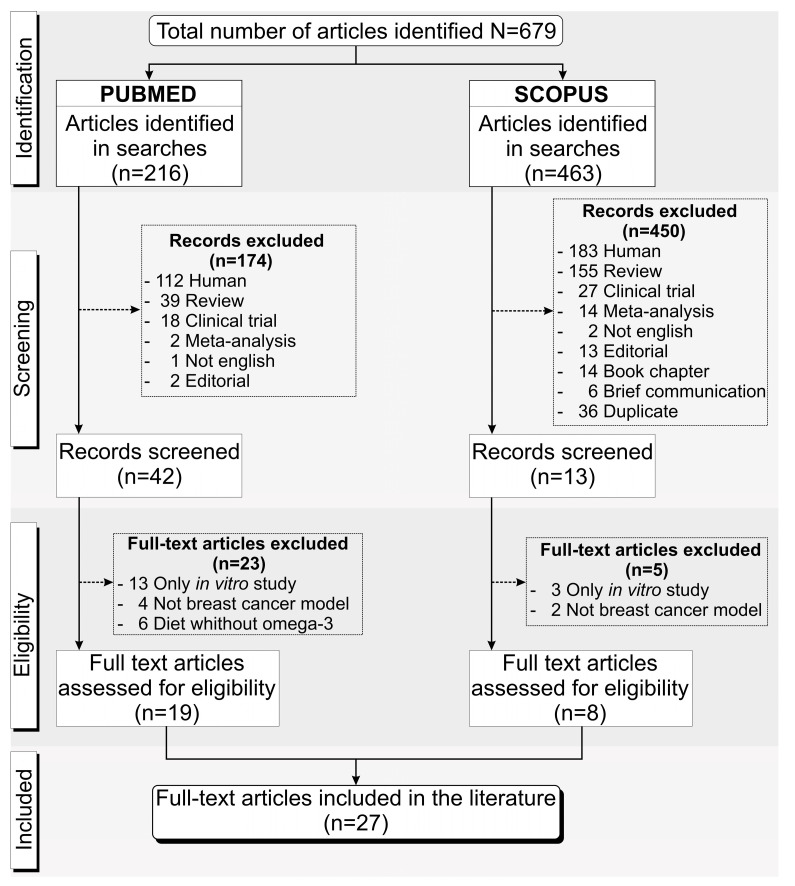
A flowchart of the systematic review, identifying at each stage of the PRISMA guidelines, the number and reasons for excluding studies until the final stage of the inclusion of studies.

**Figure 2 nutrients-15-01310-f002:**
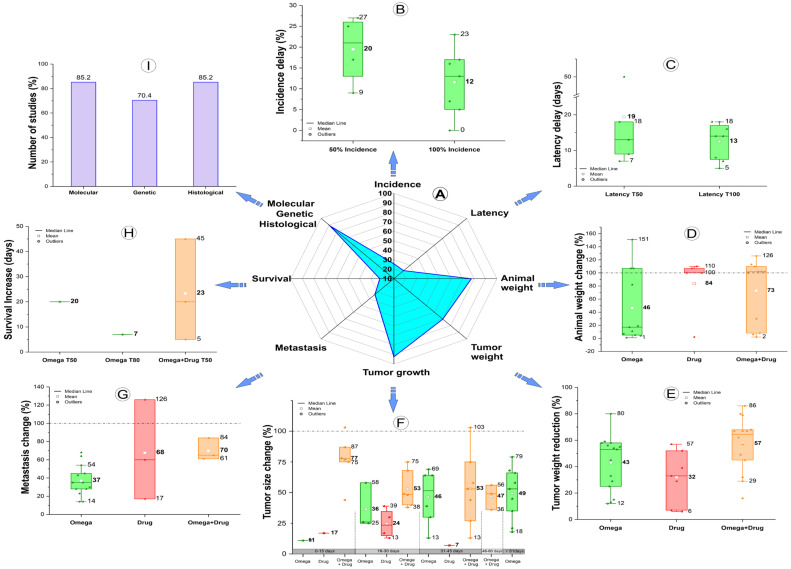
Outcomes of the omega-3 polyunsaturated fatty acid (ω-3 PUFA) treatment effects combined with antitumor drugs or not. (**A**) In the middle of the figure, a spider chart of the frequency of each aspect of tumor analysis outcomes was reported by the studies. The box plots around this chart represent the analysis of these tumor outcomes according to the experimental conditions analyzed (the ω-3 PUFA effect, antitumor drugs’ effect, or the combined treatment effect) versus their control data in each study; (**B**) tumor incidence; (**C**) tumor latency; (**D**) animal weight; (**E**) tumor weight; (**F**) tumor growth analyzed at different time points; (**G**) tumor metastasis; (**H**) survival rate; (**I**) the reporting frequency of the molecular analysis. The genetic and histological analyses are also represented by the histogram graphic.

**Figure 3 nutrients-15-01310-f003:**
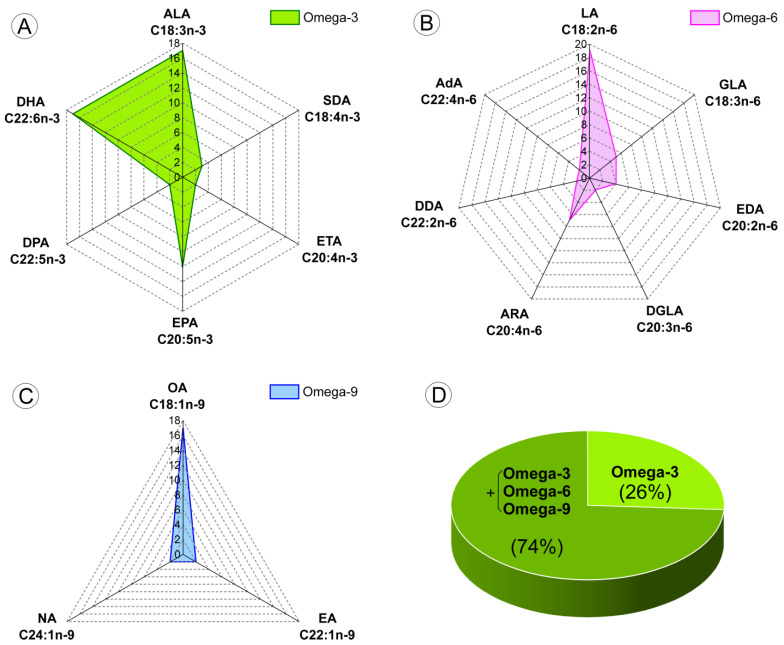
The fatty acids (FA) composition analysis according to the frequency of citation of omega-3-6-9 FAs types in the selected studies and the combined used in the breast cancer model. The frequency of use of the types of omega-3 PUFA (A)omega-6 PUFA (B), omega-9 MUFA (C), and the percentage of studies that used omega-3 alone or omega-3-6-9 combined (D).

**Table 1 nutrients-15-01310-t001:** Characteristics of the breast cancer experimental animal model.

Ref	Year	Characteristics of Animal Model		Animal Description				
		Type of Model	Source of Tumor	Specie	Strain	Genotype	Sex	Age (Week)
Newell et al. [17]	2022	Induction by cell	Human	Mice	NSG	NOD.Cg-*Prkdc^scid^Il2rg^tm1Wjl^*/SzJ	F	6
Li et al. [18]	2022	Induction by cell	Human	Mice	BALB/c (Nude)	*nu/nu*	F	3–4
Wang et al. [39]	2021	Induction by cell	Mice	Mice	BALB/c	Wildtype	F	6–7
Monk et al. [40]	2021	Transgenic	Spontaneous	Mice	FVB/N × MMTV	MMTV-*Neu^ndl^*YD5	F	4
Luo et al. [19]	2021	Induction by cell	Human	Mice	BALB/c (Nude)	J:nu	M	NR
Ion et al. [41]	2021	Transgenic	Spontaneous	Mice	SV129 × *c(3)1-TAg*	Hemizygous pups	F	3
Guo et al. [42]	2021	Induction by cell	Mice	Mice	BALB/c	Wildtype	F	8
Garay et al. [43]	2021	Induction by cell	Mice	Mice	BALB/c	Wildtype	F/M	NR
Abbas et al. [44]	2021	Induction by drug	DMBA drug	Mice	BALB/c	Wildtype	F	~7.1
Newell et al. [45]	2020	Induction by cell	Human	Mice	BALB/c (Nude)	*nu/nu*	F	6
Liu et al. [46]	2020	Induction by cell	Mice	Mice	C57BL/6	A^−^FABP^−/−^	NR	NR
					C57BL/6	Wildtype		
Hillyer et al. [47]	2020	Transgenic	Spontaneous	Mice	FVB/N × MMTV	MMTV-*Neu^ndl^*YD5	F	NR
Goupille, et al. [48]	2020	Induction by drug	NMU drug	Rats	Sprague-Dawley	Wildtype	F	6
Torres-Adorno et al. [49]	2019	Induction by cell	Human	Mice	NSG	NOD.Cg-*Prkdc^scid^Il2rg^tm1Wjl^*/SzJ	F	4–6
Newell et al. [50]	2019	Induction by cell	Human	Mice	NSG	NOD.Cg-*Prkdc^scid^Il2rg^tm1Wjl^*/SzJ	F	6
Newell et al. [51]	2019	Induction by cell	Human	Mice	BALB/c (Nude)	*nu/nu*	F	6
Li et al. [52]	2019	Induction by drug	DMBA drug	Mice	Fat-1 × C57BL/6J	Wild-genotype offspring	F	3
Fernando et al. [53]	2019	Induction by cell	Human	Mice	NSG	Wildtype	F	6–8
		Induction by cell	Mice		BALB/c	Wildtype		
Białek et al. [54]	2019	Induction by drug	DMBA drug	Rats	Sprague-Dawley	Wildtype	F	~5.3
Zhu et al. [55]	2018	Induction by cell	Human	Mice	BALB/c (Nude)	*nu/nu*	F	4–5
		Transgenic	Spontaneous		FVB/N × MMTV	FVB/N-Tg(MMTV-PyVT)634Mul		6
Liu et al. [56]	2018	Transgenic	Spontaneous	Mice	FVB/N × MMTV^+/−^	MMTV-*Neu^ndl^*YD6	F	3
Khadge et al. [57]	2018	Induction by cell	Mice	Mice	BALB/c	Wildtype	F	6
Jiao et al. [58]	2018	Induction by cell	Human	Mice	BALB/c (Nude)	*Foxn1^nu^*	F	5
Zhu et al. [59]	2017	Induction by cell	Human	Mice	BALB/c (Nude)	nu/nu	F	4–5
Vara-Messler et al. [60]	2017	Induction by cell	Human	Mice	BALB/c	Wildtype	F/M	NR
Jiao et al. [61]	2017	Induction by cell	Human	Mice	BALB/c (Nude)	*nu/nu*	F	5
Dyari et al. [62]	2017	Induction by cell	Human	Mice	BALB/c (Nude)	*nu/nu*	F	6

Abbreviations: DMBA: 7,12-Dimethylbenz(a)anthracene; NMU: n-methyl-n-nitrosourea; NSG: NOD SCID Gamma; MMTV: Mammary Tumor Virus; F: Female; M: Male; NA: Not Applicable; NR: Not Reported. ^+/−^: Heterozygous.

**Table 2 nutrients-15-01310-t002:** Breast cancer cell induction models.

Ref	Cell Characteristics			Breast Cancer Induction Tumor					
	Tumor Tissue	Cell (Tumor Subtype)	Culture Medium % FBS	Strain	Cell Number	Administration Volume (µL)	Vehicle	Local Administration	Grafts
Newell et al. [17]	Adenocarcinoma	MAXF401 (TNBC)	NR	NSG modified	NA	30 mm^3^	NA	Left flank	Xenograft
	Invasive ductal carcinoma	MAXF574 (TNBC)							
Li et al. [18]	Mammary adenocarcinoma	MCF7 (Luminal A)	DMEM 10% FBS	BALB/c (Nude)	5 × 10^5^	80	NR	Right flank	Xenograft
Wang et al. [39]	Mammary adenocarcinoma	4T1 (TNBC)	NR	BALB/c	10^7^	NR	NR	Armpit	Allograft
Luo et al. [19]	Mammary adenocarcinoma	MCF7-CSC (Luminal A)	DMEM F12 FBS-free	BALB/c (Nude) modified	1.5 × 10^7^	150	PBS	Lower right flank	Xenograft
Guo et al. [42]	Mammary adenocarcinoma	4T1 (TNBC)	RPMI 1640	BALB/c modified	10^5^	100	RPM-1640	Right hind thigh	Allograft
Garay et al. [43]	Mammary adenocarcinoma	LMM3 (TNBC)	MEM 10% FBS	BALB/c	10^6^	NR	NR	Right flank	Allograft
Newell et al. [45]	Mammary adenocarcinoma	MDA-MB-231 (TNBC)	Iscove’s MD medium 5% FCS	BALB/c (Nude)	2 × 10^6^	100	Iscove’s MD medium 5% FCS	Below the upper right scapula	Xenograft
Liu et al. [46]	Mammary adenocarcinoma	E0771 (Luminal B)	RPMI 1640 5% FBS	C57BL/6 modified	5 × 10^5^	NR	NR	Fat pad of the 4^th^ mammary gland	Allograft
		MMT060562 (TNBC)		C57BL/6					
Torres-Adorno et al. [49]	Mammary adenocarcinoma	SUM149PT (TNBC)	DMEM F12 5% FBS	NSG modified	5 × 10^5^	100	NR	Fourth inguinal mammary fat pad	Xenograft
		BCX010 (TNBC)							
Newell et al. [50]	Invasive ductal carcinoma	MAXF574 (TNBC)	NR	NSG modified	NR	NR	NR	Left flank	Xenograft
	Adenocarcinoma	MAXF401 (TNBC)							
Newell et al. [51]	Mammary adenocarcinoma	MDA-MB-231 (TNBC)	Iscove’s MD medium 5% FCS	BALB/c (Nude)	2 × 10^6^	100	Iscove’s MD medium 5% FCS	Below the upper right scapula	Xenograft
Fernando et al. [53]	Mammary adenocarcinoma	GFP-MDA-MB-231 (TNBC)	DMEM 10% FBS	NSG	2 × 10^6^	50	PBS and Matrigel (1:1)	Left inguinal mammary fat pad	Xenograft
	Mammary adenocarcinoma	4T1 (TNBC)		BALB/c	10^5^		PBS		Allograft
Zhu et al. [55]	Mammary adenocarcinoma	MDA-MB-231 (TNBC)	DMEM 10% FBS	BALB/c (Nude)	4 × 10^5^	NR	NR	NR	Xenograft
Khadge et al. [57]	Mammary adenocarcinoma	4T1 (TNBC)	DMEM 10% FBS	BALB/c	5 × 10^3^	100	CMF-HBSS	Left inguinal 5^th^ mammary fat pad	Allograft
Jiao et al. [58]	Mammary adenocarcinoma	MDA-MB-231 (TNBC)	DMEM 10% FBS	BALB/c (Nude) modified	3 × 10^6^	100	PBS with 20% Matrigel	Flanks	Xenograft
Zhu et al. [59]	Mammary adenocarcinoma	MDA-MB-231 (TNBC)	DMEM 10% FBS	BALB/c (Nude)	4 × 10^5^	NR	NR	NR	Xenograft
Vara-Messler et al. [60]	Mammary adenocarcinoma	LM3 *	MEM 10% FBS	BALB/c	2.5 × 10^5^	200	MEM	Left flank	Allograft
					5 × 10^5^				
					10^6^				
Jiao et al. [61]	Mammary adenocarcinoma	MDA-MB-231 (TNBC)	DMEM 10% FBS	BALB/c (Nude)	10^6^	100	PBS with matrigel	Left flank	Xenograft
					5 × 10^6^				
Dyari et al. [62]	Mammary adenocarcinoma	MDA-MB-231 (TNBC)	DMEM 10% FBS	BALB/c (Nude)	4 × 10^4^	100	Ice-cold PBS with Matrigel (1:1)	Left inguinal mammary gland	Xenograft

Abbreviations: Ref.: Reference; NR: Not reported; DMEM: Dulbecco’s modified Eagle medium; FBS: Fetal Bovine Serum; DMEM F12: Dulbecco’s modified Eagle medium-F12; RPMI 1640: Roswell Park Memorial Institute 1640 media culture; MEM: Modified Eagle Medium; Iscove’s MD medium: Iscove’s Modified Dulbecco’s medium; FCS: Fetal Calf Serum; NSG: NOD SCID GAMMA; PBS: Phosphate-Buffered Saline; CMF-HBSS: Ca++ Mg++ free Hank’s balanced salt solution. Note: * this cells did not show a clear classification of the BC type.

**Table 3 nutrients-15-01310-t003:** In vivo experimental design and characteristics of the diet and complementary treatment.

Ref	Groups	n/N	Diet	FAs supplement							Treatment	
				FAs Proposal	ω-3 (g/kg)	ω-6 (g/kg)	ω-9 (g/kg)	Source	Via	Time (Week)	Drug/Technique	Time
Newell et al. [17]	Control	8/32	Standard	Prevention and treatment	4.8	42.84	72.8	Oil of lard, vegetable, canola, olive, flax and Arasco	Oral	7	0.9% saline or DTX–IP	6 (2×/w)
	Low and high DHA				14.4	27/31.2	81.6/77.4	Canola Oil/DHAsco + vegetable oil			DTX–IP	
Li et al. [18]	Control	6/42	NR	Treatment	NA	NA	NA	NA	NA	Single dose	Saline or Taxol or PTX LN	Single dose
	DHA/FA/LNs		NR or FA		NR			Soybean oil, cholesterol, egg phosphatidylcholine and Croda Inc.	IV		PTX by LN– IV	
Wang et al. [39]	Control or Ce6	5/30	NR + HA	Treatment	NA	NA	NA	Innochem^®^	IV	Single dose	Saline or DTX IV or NIR	Single dose (after 1 and 24 h)
	CHD NP											
Monk et al. [40]	Low fat	~10–16/30–48	Experimental	Treatment	NR	NR	NR	Lard and corn oils	Oral	16	NA	NA
	High fat				18.24			Lard, corn, menhaden, and fish oils				
Luo et al. [19]	Control	3–4/7	Chow	Prevention and treatment	NR	NR	NR	Corn oil	Gavage	~11	NA	NA
	ω3 PUFA + AA				90 µl			NR				
Ion, et al. [41]	Control/High ω3 (mothers)	23/46	AIN-76A	Prevention and treatment	50 or 40	0 or 20	NR	Corn and canola oils or fish oil	Oral	2	NA	NA
	CC/FC	4/16			NR	50		Corn and canola oil		~17; 18.5 and 20		
	CF/FF				40	20		Fish oil				
Guo et al. [42]	Control Negative (CNB)	8/64	Standard Se free	Treatment	NA	NA	NA	NA	NA	NA	Saline	~4 (1×/every 4 days)
	Control Positive (TB)											
	TB-NS										NA	
	TB-TAX or Adr or Ava										TAX–IP	
											Adr–IV	
											Ava–IP	
	TB-NS TAX or Adr or Ava				3.52 µg	NA	NA	Nutrition supplementation (NS) with Selenium (Se)	Gavage	~4 (2×/day)	TAX–IP	
											Adr–IV	
											Ava–IP	
Garay et al. [43]	CO-diet (ω6/ω3)	8–10/24–30	AIN-93	Prevention	0.396	32.31	19,14	Corn oil	Oral	~13	NA	NA
	SO-diet (ω6)				0.014	45.48	8.56	Safflower oil				
	ChO-diet (ω3)				37.8	12.80	4.03	Chia oil				
Abbas et al. [44]	Ca (Phase I)	15/30	AIN-93	Epigenetic modulation and prevention	NR	NR	NR	Canola oil or Corn oil (mother diet)	Oral	14d	NA	NA
	Co (Phase I)											
	Ca-Co (Phase II)	100/200								50d		
	Co-Co pups (Phase II)											
	Ca-Co (Phase III)	50/200						Corn oil		42d		
	Co-Co (Phase III)											
Newell et al. [45]	OLA/LNA (Control)	6/12	AIN-76	Treatment	0	31.6	95.8	Oleic and linoleic acid	Oral	5	Saline or DOX	4 (2×/w)
	DHA				6.8	27.8	90.4	DHAsco				
Liu et al. [46]	LFD (WT × A-FABP^−/−^)	9/36	Diet 10% fat	Treatment	NR	NR	NR	Soybean oil	Oral	5 months	NA	NA
	HFD-C		Diet 45% fat		0			Cocoa butter, soybean oils				
	HFD-F (WT × A-FABP^−/−^)				175.5			Fish and soybean oils				
Hillyer et al. [47]	ω6 PUFA, control	11/46	Standard	Prevention and treatment	0.49	72.09	15.44	Safflower oil	Oral	20	NA	NA
	ω3 PUFA, control	10/46			9.99	52.99	14.1	Menhaden and safflower oils				
	ω3 PUFA	9/46			17.44	53.90	16.76	Flaxseed and safflower oils				
	MUFA	6/46			0.66	11.15	64.72	Olive oil				
	SFA	10/46			2.1	22.46	35.78	Lard				
Goupille, et al. [48]	Control	14/56	Standard	Prevention and treatment	0	NR	NR	Peanut and rapeseed oils	Oral	9	DTX–IP	6 (1×/w)
	ω3 LCPUFA		Enriched diet		35	NR	NR	Peanut, rapeseed and fish oils				
Torres-Adorno et al. [49]	Control (SUM149PT × BCX010)	9–10/27–30	AIN-76A	Treatment	NA	NA	NA	NA	Oral	~6	Dasatinib–IP	~6 (6×/w)
	EPA (SUM149PT × BCX010)				0.4 or 0.8			Fish oil				
Newell et al. [50]	Control	8/32	NR	Prevention and treatment	NA	52.46	69.8	NR	Oral	7	Saline or DTX	6 (2×/w)
	DHA				7.8	43.06	67.2	DHAsco				
Newell et al. [51]	Control	6/24	AIN-76	Prevention and treatment	NR	30.98	95.8	Sunflower, canola, olive, and ARAsco oils	Oral	7	Saline or DOX	6 (2×/w)
	DHA				6.76	27.64	90.4	DHAsco				
Li et al. [52]	Control	24/43	AIN-93G	Prevention	NR	NR	NR	Soybean oil	Oral	3–7	NA	NA
	Fat-1	19/43						ω3 PUFA endogenous from soybean oil				
Fernando et al. [53]	Control (4T1 × MDA-MB-231)	NR	Standard	Treatment	NA	NA	NA	NA	IP	5× in 9 days and 20× in 39 days	Saline	5× in 9 days and 20× in 39 days
	PZ-DHA (4T1 × MDA-MB-231)							NR			PZ	
Białek et al. [54]	SAF (control)	8/46	Standard or Labofeed H	Prevention and treatment	0.95	75.3	130.24	Safflower oil	Gavage	21	NA	NA
	SAF-plus	14/46										
	CLA (control)	7/46			NA	42.7	37.187	Bio conjugated linoleic acid				
	CLA-plus	17/46										
Zhu et al. [55]	Cell induction model–Control, EPA, RP and EPA-RP	10/40	Standard	Treatment	30	NA	NA	EPA commercial product	Oral	2	Saline or Rapamycin	2
	Transgenic model–Control, EPA, RP, EPA-RP and EPA-RP-NAC	6/30						EPA commercial product or NAC		4		4
Liu et al. [56]	Control (ω6)	~8/32	AIN-93G	Prevention and treatment	0.422	15.796	3.586	Safflower oil	Oral	6 and 20	NA	NA
	FS (ω3)				1.14	3.66963	1.056	Flaxseed oil				
	FS-SF (ω3e6)				12.694	3.63	3.3	Flaxseed and safflower oil				
	Menh-SF (ω3 > 6)				0.43164	3.7686	0.9504	Menhaden fish and safflower oil				
Khadge et al. [57]	ω6 (control)	20/40	Lieber-DeCarli (control diet)	Prevention	NR	NR	NR	Corn, olive and safflower oils	Oral	10–16	NA	NA
	ω3		Lieber-DeCarli modified					Corn, olive, fish and safflower oils				
Jiao et al. [58]	ω3	5/20	NR	Treatment	NR	NR	NR	Fish oil	Oral	Throughout	Vehicle or SFN–IP	Every 2 days/9 days
	ω6							Corn oil				
Zhu et al. [59]	Control	10/40	NR	Treatment	30	NA	NA	EPA commercial product	NR	2	Vehicle or RA	2
	RA (retinoic acid)											
	EPA											
	RA and EPA											
Vara-Messler et al. [60]	CO-diet (ω6)	20/37	Standard	Treatment	0.8686	22.7943	5.7749	Corn oil	Oral	~6	NA	NA
	ChO-diet (ω3)	17/37			27.1932	10.1695	0.9847	Chia seed oil				
Jiao et al. [61]	Less aggressive model	5/40	Standard	Prevention and treatment	NR	NR	NR	Corn oil	Oral	Throughout	Vehicle or DSF–IP	5 days
								Fish oil				
	More aggressive model							Corn oil				
								Fish oil				
Dyari et al. [62]	Control	~4–8/24	NR	Treatment	NR	NR	NR	Corn oil	NA	6 (1×/day)	Vehicle or AUDA–IP	6 (1×/day)
	AUDA (vehicle)								IP			
	C20E 0.05							ω3 endogenous from corn oil	IP			
	C20E 0.5											

Abbreviations: Ref: Reference; n/N: Animals per group/Animals total; FAs: Fatty Acids; ω-3: Omega-3; ω-6: Omega-6; ω-9: Omega-9; DHA: Docosahexaenoic acid; DTX: Docetaxel; IP: Intraperitoneal; FA: Folic Acid; LNs: Lipid Nanoemulsions; NR: Not Reported; NA: Not Applicable; IV: Intravenous; PTX: Paclitaxel; Ce6: Chlorin e6; CHD: Cys-DHA/Ce6; NP: Nanoparticles; HA: Hyaluronic Acid; AA: Arachidonic Acid; CC: Corn/Corn; FC: Fish/Corn CF: Corn/Fish; FF: Fish/Fish CNB: Negative Control; TB: Positive Control; NS: Nutritional Supplements: TAX: Taxol; Adr: Adriamycin; Ava: Avastin; Se: Selenium; CO: Corn Oil; SO: Safflower Oil; ChO: Chia Oil; Ca: Canola Oil: Co: Corn Oil; OLA: Oleic Acid; LNA: Linoleic Acid; DOX: Doxorubicin; LFD: Low-Fat Diet: WT: Wild-Type: A-FABP: A-FABP–Deficient Mice; HFD: High-Fat Diet; HFD-C: High-Fat Diets (cocoa butter); HFD-F: High-Fat Diets (fish oil); PUFA: Polyunsaturated Fatty Acids; MUFA: Monounsaturated Fatty Acids; SFA: Saturated Fatty Acids; LCPUFA: Long Chain Polyunsaturated Fatty Acids; SUM149PT: Xenograft Tumor Model of TNBC; TNBC: Triple-Negative Breast Cancer; BCX010: TNBC Cell Line; EPA: Eicosapentaenoic acid; FAT-1: Transgenic fat-1 Mouse Model; 4T1: Breast Cancer Cell Model; MDA-MB-231: TNBC Cell model; PZ: Phloridzin; SAF: Safflower; CLA: Conjugated Linoleic Acid; RP: Rapamycin; NAC: N-acetyl-L-cysteine; FS: Flaxseed; SF: Safflower; Menh: Menhaden; SFN: Sorafenib; RA: Retinoic Acid; DSF: Disulfiram; AUDA: 12-(3adamantan-1-yl-ureido)-Dodecanoic Acid; C20E: ω3-eicosapentaenoic acid.

**Table 4 nutrients-15-01310-t004:** Description of FAs composition used in the selected studies in the breast cancer model.

Ref	Groups	ω-3 PUFA							ω-6 PUFA								ω-9 MUFA			
		Total	ALA C18:3	SDA C18:4	ETA C20:3	EPA C20:5	DPA C22:5	DHA C22:6	Total	LA C18:2	GLA C18:3	EDA 20:2	DGLA C20:3	ARA C20:4	DDA C22:2	AdA C22:4	Total	OA C18:1	EA C22:1	NerA C24:1
Newell et al. [17]	Control	4.8 g/kg	4.8 g/kg						42.8 g/kg	42 g/kg				0.84 g/kg			72.8 g/kg	72.8 g/kg		
	High DHA	14.4 g/kg	6.8 g/kg					7.6 g/kg	31.2 g/kg	30.4 g/kg				0.8 g/kg			77.4 g/kg	77.4 g/kg		
	Low DHA	14.4 g/kg	10.4 g/kg	0.8 g/kg				3.2 g/kg	27.0 g/kg	26.2 g/kg							81.6 g/kg	81.6 g/kg		
Li et al. [18]	DHA/FA/LN	30 μL/mL						30 μL/mL	NA								NA			
Wang et al. [39]	CHD NP	NA						+	NA								NA			
Monk et al. [40]	High fat	18.24 g/kg	+			+		+	NR	+							NR	+		
Luo et al. [19]	ω3 PUFA	90 ul	+			+		+	NR	+				+			NR			
Ion, et al. [41]	Control ω3								50 g/kg	+		+					NR			
	(mothers)	40 g/kg	+			+		+	20 g/kg	+		+					NR			
	CC/FC								50 g/kg	+		+					NR			
	CF/FF	40 g/kg	+			+		+	20 g/kg	+		+					NR			
Guo et al. [42]	Control	NA				NA		NA	NA								NA			
	TB-NS	3.52 ug				2.04 ug		1.48 ug	NA								NA			
	TB-NS drugs																			
Garay et al. [43]	ChO-diet (ω3)	37.8 g/kg	37.8 g/kg						12.8 g/kg	12.8 g/kg				+			4.03 g/kg	4.03 g/kg		
	CO-diet (ω6/ω3)	0.4 g/kg	0.4 g/kg						32.3 g/kg	32.3 g/kg				+			19.1 g/kg	19.1 g/kg		
	SO-diet (ω6)	0.014 g/kg	0.014 g/kg						45.5 g/kg	45.5 g/kg				+			8.56 g/kg	8.56 g/kg		
Abbas et al. [44]	Ca(Phase I)	NR	+						NR	+							NR	+		
	Co(Phase I)	NR	+						NR	+							NR	+		
	Ca/Co(Phase II)	NR	+						NR	+							NR	+		
	Co/Co(Phase II)	NR	+						NR	+							NR	+		
	Ca/Co(Phase III)	NR	+						NR	+							NR	+		
	Co/Co(Phase III)	NR	+						NR	+							NR	+		
Newell M et al. [45]	Control	0							31.6 g/kg	27.8 g/kg	2.26 g/kg			1.0 g/kg			95.8 g/kg	95.8 g/kg		
	DHA	6.8 g/kg	1.2 g/kg					5.6 g/kg	27.8 g/kg	26.2 g/kg	0.6 g/kg			1.0 g/kg			90.4 g/kg	90.4 g/kg		
Liu et al. [46]	HFD-C	0 g/kg							NR								NR			
	HFD-F	175.5 g/kg				+		+	NR								NR			
Hillyer et al. [47]	ω6 PUFA, control	0.49 g/kg	0.31 g/kg	0.13 g/kg	0 g/kg	0.05 g/kg			72.1 g/kg	71.93 g/kg	0.11 g/kg	0.05 g/kg				0.11 g/kg	15.4 g/kg	15.25 g/kg	0.02 g/kg	0.17 g/kg
	ω3 PUFA, control	9.99 g/kg	0.86 g/kg	0.9 g/kg	0.2 g/kg	4.20 g/kg	0.75 g/kg	3.08 g/kg	53 g/kg	51.7 g/kg	0.16 g/kg	0.12 g/kg	0.10 g/kg	0.49 g/kg	0.27 g/kg	0.14 g/kg	14.1 g/kg	13.78 g/kg	0.09 g/kg	0.2 g/kg
	ω3 PUFA	17.44 g/kg	17.3 g/kg	0.1 g/kg					53.9 g/kg	53.81 g/kg		0.09 g/kg					16.8 g/kg	16.59 g/kg	0.02 g/kg	0.2 g/kg
	MUFA	0.66 g/kg	0.66 g/kg						11.2 g/kg	11.2 g/kg							64.7 g/kg	64.50 g/kg	0.03 g/kg	0.2 g/kg
	SFA	2.1 g/kg	1.56 g/kg	0.2 g/kg	0.3 g/kg		0.12g/kg		22.5 g/kg	20.85 g/kg	0.11 g/kg	0.84 g/kg	0.17 g/kg	0.37 g/kg		0.12 g/kg	35.8 g/kg	35.76 g/kg	0.02 g/kg	
Goupille et al. [48]	Control	0 g/kg							NR	+							NR	+		
	ω3 LCPUFA	35 g/kg				10 g/kg		25 g/kg	NR	+							NR	+		
Torres-Adorno et al. [49]	EPA	0.4 g/kg				0.4 g/kg			NA								NA			
	EPA	0.8 g/kg				0.8 g/Kg			NA								NA			
Newell et al. [51]	Control	NA							52.5 g/kg	47.8 g/kg	3.8 g/kg	0.86 g/kg					69.8 g/kg	69.8 g/kg		
	DHA	7.8 g/kg						7.8 g/kg	43.1 g/kg	37 g/kg	5.2 g/kg	0.86 g/kg					67.2 g/kg	67.2 g/kg		
Newell et al. [50]	Control	NR							31 g/kg	27.8 g/kg	2.26 g/kg			0.92 g/kg			95.8 g/kg	95.8 g/kg		
	DHA	6.76 g/kg	1.12 g/kg					5.64 g/kg	27.6 g/kg	26.2 g/kg	0.52 g/kg			0.92 g/kg			90.4 g/kg	90.4 g/kg		
Li et al. [52]	Control								NR	+							NR	+		
	Fat-1	NR	+						NR	+							NR	+		
Fernando et al. [53]	Control	NR							NR								NR			
	PZ-DHA	NA						+	NA								NA			
Białek et al. [54]	SAF (control)	0.95 g/kg	0.95 g/kg						75.3 g/kg	75.3 g/kg							130.2 g/kg	130 g/kg		0.24 g/kg
	SAF-plus	0.95 g/kg	0.95 g/kg						75.3 g/kg	75.3 g/kg							130.2 g/kg	130 g/kg		0.24 g/kg
	CLA (control)								42.7 g/kg	42.7 g/kg							37.2 g/kg	37 g/kg		0.2 g/kg
	CLA-plus								42.7 g/kg	42.7 g/kg							37.2 g/kg	37 g/kg		0.2 g/kg
Zhu et al. [55]	Cell induction model- EPA	30 g/kg				30g/kg			NA								NA			
	Transgenic model- EPA	30 g/kg				30g/kg			NA								NA			
Liu et al. [56]	Control (ω6)	0.422 g/kg	0.04 g/kg	0.02 g/kg	-	-	-	-	15.8 g/kg	15.73 g/kg	0.07 g/kg						3.59 g/kg	3.59 g/kg		
	FS (ω3)	1.14 g/kg	1.14 g/kg	-	-	-	-	-	3.67 g/kg	3.66 g/kg	0.01 g/kg						1.06 g/kg	1.06 g/kg		
	FS-SF (ω3:w6)	12.7 g/kg	12.7 g/kg	-	0.02 g/kg	-	-	-	3.63 g/kg	3.63 g/kg	-						3.3 g/kg	3.3 g/kg		
	Menh-SF (ω3>w6)	0.43 g/kg	0.04 g/kg	0.05 g/kg	0.01 g/kg	0.26 g/kg	0.05 g/kg	0.02 g/kg	3.8 g/kg	3.7 g/kg	0.01 g/kg	0.01 g/kg	0.01 g/kg	0.03 g/kg	0.01 g/kg	0.01 g/kg	0.95 g/kg	0.94 g/kg	0.01 g/kg	
Khadge et al. [57]	ω6 (control)	NR	+						NR	+							NR	+		
	ω3	NR	+						NR	+							NR	+		
Jiao et al. [58]	ω-6	NR	+						NR	+							NR	+		
	ω-3	NR				+		+	NR								NR			
Zhu et al. [59]	EPA	30 g/kg	+			30 g/kg		+	NA								NA			
Vara-Messler et al. [60]	CO-diet (ω6)	27.2 g/kg	27.2 g/kg						10.2 g/kg	10.2 g/kg							0.98 g/kg	0.98 g/kg		
	ChO-diet (ω3)	0.87 g/kg	0.87 g/kg						22.8 g/kg	22.8 g/kg							5.8 g/kg	5.8 g/kg		
Jiao et al. [61]	Less aggressive model	NR	+						NR	+							NR	+		
	More aggressive model	NR	+						NR								NR			
Dyari et al. [62]	C20E	NR	+						NR	+							NR	+		

Abbreviations: Ref: Reference; FAs: Fatty Acids; ALA: Alpha Linolenic Acid; SDA: Stearidonic Acid; ETA: Eicosatrienoic Acid; EPA: Eicosapentaenoic Acid; DPA: Docosapentaenoic Acid; DHA: Docosahexaenoic Acid; LA: Linoleic Acid; GLA: Gamma-Linolenic Acid; EDA: Eicosadienoic Acid; DGLA: Dihomo-Gamma-Linolenic Acid; ARA: Arachidonic Acid; DDA: Docosadienoic Acid; AdA: Docosatetraenoic Acid; OA: Oleic Acid; EA: Erucic Acid; NerA: Nervonic Acid; FA: Folic Acid; LNs: Lipid Nanoemulsions; NA: Not Applicable; CHD: Cys-DHA/Ce6; NP: Nanoparticles; +: Positive/Used; NR: Not Reported; ω-3: Omega-3; PUFA: PUFA: Polyunsaturated Fatty Acids; CC: Corn/Corn; CF: Corn/Fish; FC: Fish/Corn; FF: Fish/Fish; TB: Positive Control; NS: Nutritional Supplements; ChO: Chia Oil; CO or Co: Corn Oil; SO or SAF or SF: Safflower Oil; Ca: Canola Oil; HFD-C: High-Fat Diets (cocoa butter); HFD-F: High-Fat Diets (fish oil); ω-6: Omega-6; ω-9: Omega-9; PUFA: Polyunsaturated Fatty Acids; MUFA: Monounsaturated Fatty Acids; SFA: Saturated Fatty Acids; LCPUFA: Long Chain Polyunsaturated Fatty Acids; Fat-1: Transgenic fat-1 Mouse Model; Plus: DMBA; DMBA: 7,12-dimetilbenz(a)antraceno; CLA: Conjugated Linoleic Acid; FS: Flaxseed; Menh: Menhaden; C20E: ω3-eicosapentaenoic acid.

## Data Availability

Not applicable.

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
