# Peer review of "The Effects of Omega-3 Polyunsaturated Fatty Acids on Breast Cancer as a Preventive Measure or as an Adjunct to Conventional Treatments"

_nutrients, 2023, doi:10.3390/nu15061310_

Round 1

Reviewer 1 Report

The review is of relevance to shed light on the importance of omega-3 fatty acids for the successful treatment of BC. However, there are some editorial problems with the manuscript and some points that require revision and clarification. The title should be clear about that king of omega-3 compounds are being considered, on the first line of the abstract it is written that, in fact, the compounds are “omega-3 FAs”. The authors should not use the “fatty acids” the first time the compounds are referred.

The sentences often lack precision and some need to be re-written to clarify the information. I have made multiple comments and amendments throughout the text, especially in the introduction (on the pdf file), and below in the detailed comments.

The authors, in the abstract refer that their analysis is based in five (5) topics, among which is “breast cancer model” used (line 18) and go on to say that “There are diverse and well-established models in the literature on breast cancer” (line 20). I suppose that the BC models are established in the clinical practice and referred in scientific literature. In fact there are several types or categories of BC, not necessarily “models”; this should be revised. There is a certain confusion regarding the BC “model” which is the object of the study: Until line 88, the reader is led to believe that the review is on the use of Omega-3 fatty acids during BC cancer treatments and only in line 88 it is revealed that the revision concerns studies using animal models of BC. This should be made clear in the abstract.

The confusion with the models and types of BC is again present in line 131 with “different types of breast tumor models”. At this point the reader assumes the search was for “animal models” (line 88), which is repeated in line 202, although that is nor specified in the “Search Strategy”. Do the authors mean (line 131) animal BC models of the different BC types (classified according the receptor presence)? Or do they mean the cell transplants? Please note that only in line 202 does the reader learn of “tumor cell transplant”.

I consider that the description of the mice strains could be shortened since much of the information is repeated in table 1. The information of the BC cell lines (MCF-7, MDA-MB-231, etc) used doesn’t refer to what type of BC they are. This is a bit odd after the explanation of BC types (e.g. positive, Her2 positive, triple negative) in the introduction.

I find figure 2 very confusing, perhaps it should be best to split it into 2 figures and revise the legend accordingly, referring each particular panel.

I consider that the authors dedicate much of the review to the details of the analysed studies somewhat neglecting to highlight the important results of those studies, namely in addressing the title of the manuscript “Effect of Omega-3 fatty acids on Breast Cancer as a Preventive Measure or as an Adjunct to Conventional Treatments”.

The text should also be revised to avoid sentences such as “FAs (…) we observed that some types more frequently cited than others”. I think the point that must be made is which FAs are more used in the studies; the frequency of citation is not a very good parameter to convey their relevance in the studies.

The authors also start by calling the unsaturated fatty acids “omega” thus referring to first unsaturation from the last Carbon of the chain. Once the option is made, all further reference in the text should be to the “omega”, I don’t consider it appropriate to introduce “n” afterwards; you should change the “n-“ notation to e.g. 18:3n3 to 18:3w3. In fact, the notation w3 PUFA; w6 PUFA or w9 FA instead of e.g. “omega-3 FAs” is preferred.

It would be very interesting to learn the effect of w3 PUFA or other PUFA in relation to each BC type (either BC cell lines, transgenic or drug induced BC).

Author Response

Reviewer #1

The review is of relevance to shed light on the importance of omega-3 fatty acids for the successful treatment of BC. However, there are some editorial problems with the manuscript and some points that require revision and clarification. The title should be clear about that king of omega-3 compounds are being considered, on the first line of the abstract it is written that, in fact, the compounds are “omega-3 FAs”. The authors should not use the “fatty acids” the first time the compounds are referred.

Answer:  Thank you for your suggestion. We changed the title according to the reviewer suggest and corrected the first-time mention about ômega-3 without the ‘fatty acids’.

The sentences often lack precision and some need to be re-written to clarify the information. I have made multiple comments and amendments throughout the text, especially in the introduction (on the pdf file), and below in the detailed comments.

Answer: Thank you for your comments and suggestions. We re-written and corrected the texts highlighted in the pdf file, improving the clarity of information.

The authors, in the abstract refer that their analysis is based in five (5) topics, among which is “breast cancer model” used (line 18) and go on to say that “There are diverse and well-established models in the literature on breast cancer” (line 20). I suppose that the BC models are established in the clinical practice and referred in scientific literature. In fact there are several types or categories of BC, not necessarily “models”; this should be revised. There is a certain confusion regarding the BC “model” which is the object of the study: Until line 88, the reader is led to believe that the review is on the use of Omega-3 fatty acids during BC cancer treatments and only in line 88 it is revealed that the revision concerns studies using animal models of BC. This should be made clear in the abstract.

Answer:  Thank you for your comments. We corrected the phrases of abstract and introduction, making it clear that the studies included are preclinical research with animal models on breast cancer.

The confusion with the models and types of BC is again present in line 131 with “different types of breast tumor models”. At this point the reader assumes the search was for “animal models” (line 88), which is repeated in line 202, although that is nor specified in the “Search Strategy”. Do the authors mean (line 131) animal BC models of the different BC types (classified according the receptor presence)? Or do they mean the cell transplants? Please note that only in line 202 does the reader learn of “tumor cell transplant”.

Answer:  Thank you for your observation. We corrected confusion present in line 131 (item 2.2), we clarified that was included in the review all different types of BC induction in animal models, such as transgenic animals’ model, induction by transplantation of tumor cells or induction by drugs. These different types of models are not specified in the search strategy, but in the eligibility analysis was considered this characteristics, excluding clinical and in vitro studies, as well as the studies that did not apply the analysis in animal model or the use of omega-3. So, the search strategy question, we considered only the experimental articles in BC animal model (pre-clinical), excluded all clinical trials, and humans of the booleans, as also shown in the Figure 1. 

I consider that the description of the mice strains could be shortened since much of the information is repeated in table 1. The information of the BC cell lines (MCF-7, MDA-MB-231, etc) used doesn’t refer to what type of BC they are. This is a bit odd after the explanation of BC types (e.g. positive, Her2 positive, triple negative) in the introduction.

Answer:  Thank you for your comments. The purpose of repeating the information on the strain of the animal in Table 2 was to maintain a line of reasoning in relation to the entire process of induction of BC by implantation of tumor cells. In the first columns of Table 2, the characteristics of the tumor cells implanted and then the characteristics of breast cancer induction by the host of the cells, and the implantation process.

Furthermore, in Table 2, we added the type of BC of cells used in each study according to the status of ER, PR, HER2, breast cancer (luminal A, luminal B, HER2 positive, and triple-negative), according to the explanation did in the introduction section of the manuscript. Clarifying in the manuscript the difference between the type of BC and type of induction model of BC (transgenic model, induction by tumor cell or by drug).

I find figure 2 very confusing, perhaps it should be best to split it into 2 figures and revise the legend accordingly, referring each particular panel.

Answer:  Thank you for your comments. Changes were made to the figures to make the information clearer. The central figure (spider chart) is related to the figures that are located around it, where these have been identified by letters (B-I) for better understanding.

I consider that the authors dedicate much of the review to the details of the analysed studies somewhat neglecting to highlight the important results of those studies, namely in addressing the title of the manuscript “Effect of Omega-3 fatty acids on Breast Cancer as a Preventive Measure or as an Adjunct to Conventional Treatments”.

Answer:  We appreciate your comment, but the title of the study is related to the objective of the review, which is to verify the effect of omega-3 fatty acids in the prevention and/or treatment of breast cancer in preclinical studies. Although some results were more notable, a title emphasizing this finding distracts from the original purpose of the review. The discussion and conclusion were rewritten, making the findings clearer and in line with the title of the manuscript.

The text should also be revised to avoid sentences such as “FAs (…) we observed that some types more frequently cited than others”. I think the point that must be made is which FAs are more used in the studies; the frequency of citation is not a very good parameter to convey their relevance in the studies.

Answer:  Thank you for your observation. We corrected this citation  “FAs” in all the manuscripts, maintaining this term only for the mention that involves the different types of fatty acids.

The authors also start by calling the unsaturated fatty acids “omega” thus referring to first unsaturation from the last Carbon of the chain. Once the option is made, all further reference in the text should be to the “omega”, I don’t consider it appropriate to introduce “n” afterwards; you should change the “n-“ notation to e.g. 18:3n3 to 18:3w3. In fact, the notation w3 PUFA; w6 PUFA or w9 FA instead of e.g. “omega-3 FAs” is preferred.

Answer:  Thank you for your suggestion, we consider replacing the notation “n-“ or “omega-3 FAs” in the case of w-3 PUFA; w-6 PUFA, or w-9 FA, however, we maintain the nomenclature of 18:n3n, because although the “n” refers to the position of the last unsaturation in the carbon chain, the use of “n” instead of “w” is much more usual when referring to this molecular formula of the unsaturated fatty acids, as reported in the literature [1,2]

It would be very interesting to learn the effect of w3 PUFA or other PUFA in relation to each BC type (either BC cell lines, transgenic or drug induced BC).

Answer:  Thank you for your suggestion. The transgenic BC model and the BC model chemical-induced were reported in a few studies, but we highlighted in the results, discussion, and conclusion, the main outcomes involved in these models and the omega-3 effect condition isolated or combined with antitumor drugs.

All requested changes were made in the PDF file that was attached by reviewer 1.

Reviewer 2 Report

The authors of the article explore the effect of omega-3 on breast cancer as an adjunct to the conventional therapy and its potential in prevention. The manuscript's authors attempt to shine a light on a topic whic is gaining interest among other researchers. The article is comprehensive, therefore I will address each section separately.

Introduction

It consists well gathered information necessary to understand the rest of the article. It is logically arranged and should be accessible for any interested reader. My minor suggestions are:

- providing gene names in italics (refers to the whole article)
- Line 75 contains a mistake 'its have'. Additionally the second sentence in this paragraph is too complex - it is not clear what does the beginning refer to and while reading it is easy to get lost. I suggest spliting the long sentence into a few shorter ones.

Methods

The section is thoroughly prepared, every single aspect of the study is carefully described providing high quality of the article.

Results

Reviewed articles are meticulously described, the tables give a good simplified overview of the source articles.
I congratulate the authors on choosing spider charts for explaining specific characteristics of the studies. 

Discussion

The Discussion is extensive and rather lengthy, however since the Conclusions section is quite short and deficient in specific information, I suggest shortening Discussion and presenting the main conclusions in the first paragraph of the section OR expanding the Conclusions and providing more detailed information there - specially the numbers You consider most informative. 

Currently the key information in Discusion is somewhat scattered in several paragraphs which makes it difficult to focus on the main ideas of the article. 

Additionally - lines 640-646 - lack reference and it is unclear to what does this paragraph address.

Conclusions

From my point of view the statement "there is a positive effect of the additional use of foods mainly consisting of omega-3 FAs or its components in the prevention of BC" is vague and not very informative. I suggest providing numbers or more detailed conclusions. Moreover, it should be specified that reviewed articles are animal-based studies.

General comments

- the article gives a broad overview on the topic with some interesting conclusions, which should be a starting point in searching for the dose of FAs which can provide the desired effect

- the English language requires a revision - was it checked by a native speaker or a linguist? The grammar and the use of some words (eg. already or current) is odd in several sentences - I suggest English language editing before submitting revised manuscript.   

- I congratulate the authors on the difficult job done on creating this manuscript and I encourage applying some minor changes.

Author Response

Reviewer #2

The authors of the article explore the effect of omega-3 on breast cancer as an adjunct to the conventional therapy and its potential in prevention. The manuscript's authors attempt to shine a light on a topic whic is gaining interest among other researchers. The article is comprehensive, therefore I will address each section separately.

Introduction

It consists well gathered information necessary to understand the rest of the article. It is logically arranged and should be accessible for any interested reader. My minor suggestions are:

- providing gene names in italics (refers to the whole article)

- Line 75 contains a mistake 'its have'. Additionally the second sentence in this paragraph is too complex - it is not clear what does the beginning refer to and while reading it is easy to get lost. I suggest spliting the long sentence into a few shorter ones.

Answer:  Thank you for your observation. We corrected all mistakes, including in Line 75 appointed and splatted the long sentence into shorter ones, making it clearer and more comprehensible.   

Methods

The section is thoroughly prepared, every single aspect of the study is carefully described providing high quality of the article.

Answer:  We appreciate your commentary and dedication into this review.

Results

Reviewed articles are meticulously described, the tables give a good simplified overview of the source articles.. I congratulate the authors on choosing spider charts for explaining specific characteristics of the studies. 

Answer:  Thank you for your observation, we tried to carry out an in-depth analysis of the articles by summarizing the main results in graphs.

Discussion

The Discussion is extensive and rather lengthy, however since the Conclusions section is quite short and deficient in specific information, I suggest shortening Discussion and presenting the main conclusions in the first paragraph of the section OR expanding the Conclusions and providing more detailed information there - specially the numbers You consider most informative. 

Currently the key information in Discusion is somewhat scattered in several paragraphs which makes it difficult to focus on the main ideas of the article. 

Answer:  Thank you for your comment and suggestion. The discussion was reorganized into subtopics, to improve the clarity of the information discussed, highlighting the main conclusions in the first paragraph and the studies outcomes were discussed according to the experimental condition analyzed (omega-3 effect, antitumor drug effect, or both combined effect) and the BC model applied.

Additionally - lines 640-646 - lack reference and it is unclear to what does this paragraph address.

Answer:  Thank you for your suggestion, we added the references that were lacking and reformulated the paragraph between lines 640-646, turning it clearer and more comprehensible.

Conclusions

From my point of view the statement "there is a positive effect of the additional use of foods mainly consisting of omega-3 FAs or its components in the prevention of BC" is vague and not very informative. I suggest providing numbers or more detailed conclusions. Moreover, it should be specified that reviewed articles are animal-based studies.

Answer:  Thank you for your suggestion. The conclusion was redone and added the main conclusions considering the animal BC model and the experimental condition regarding the omega-3 isolated effect or combined with antitumor drugs.

General comments

- the article gives a broad overview on the topic with some interesting conclusions, which should be a starting point in searching for the dose of FAs which can provide the desired effect

- the English language requires a revision - was it checked by a native speaker or a linguist? The grammar and the use of some words (eg. already or current) is odd in several sentences - I suggest English language editing before submitting revised manuscript.   

- I congratulate the authors on the difficult job done on creating this manuscript and I encourage applying some minor changes.

Answer:  Thank you for all suggestions, the manuscript was reviewed by the language editing system recommended by the MDPI and in attached the certificate of service done.

Round 2

Reviewer 1 Report

I thank the authors for addressing my comments and I’m happy with the way the text was revised to incorporate my suggestions. The review is now clear in its objectives and discussion of revised studies and is of importance for better understand the importance of PUFA  as adjunct to conventional cancer treatments. I particularly liked the conclusion remarks.

Please revise the words “in vivo”, which should be in the italic form in vivo.

It is the authors’ option to maintain the n-3 or n-6 notation for ω3 or ω6 FA, respectively. I accept that although I consider that it is more accurate to use the ω notation since it refers to the position of the last unsaturation counting from the last carbon, as opposed from the first carbon (α). The use of the ω notation is itself a shorthand relative to the official nomenclature of lipids (see https://www.lipidmaps.org/lmsd_search/10103). 

The first paragraph of the discussion is made of one rather long sentence. Please break it into two sentences to improve reading.

Line 528

“…” models, and less frequently (44.4%) [19,40,41,43,44,46,47,52,54,56,57,60], but not less relevance (…)

May be changed to:

(…) models. Less frequently (44.4%) [19,40,41,43,44,46,47,52,54,56,57,60], although with relevance, (…)

Author Response

Reviewer #1

I thank the authors for addressing my comments and I’m happy with the way the text was revised to incorporate my suggestions. The review is now clear in its objectives and discussion of revised studies and is of importance for better understand the importance of PUFA  as adjunct to conventional cancer treatments. I particularly liked the conclusion remarks.

Please revise the words “in vivo”, which should be in the italic form “in vivo”.

Answer:  Thank you for your observation. We changed all “in vivo” and “in vitro” words to the italic form.

It is the authors’ option to maintain the n-3 or n-6 notation for ω3 or ω6 FA, respectively. I accept that although I consider that it is more accurate to use the ω notation since it refers to the position of the last unsaturation counting from the last carbon, as opposed from the first carbon (α). The use of the ω notation is itself a shorthand relative to the official nomenclature of lipids (see https://www.lipidmaps.org/lmsd_search/10103). 

Answer:  Thanks for your explanation, but we will keep the n-3 or n-6 notation, which is also used in the literature.

The first paragraph of the discussion is made of one rather long sentence. Please break it into two sentences to improve reading.

Line 528

“…” models, and less frequently (44.4%) [19,40,41,43,44,46,47,52,54,56,57,60], but not less relevance (…)

May be changed to:

(…) models. Less frequently (44.4%) [19,40,41,43,44,46,47,52,54,56,57,60], although with relevance, (…)

Answer:  Thank you for all comments, and we change the first paragraph of the discussion according to your suggestion.
